# An ER phospholipid hydrolase drives ER-associated mitochondrial constriction for fission and fusion

Tricia T Nguyen[1,2], Gia K Voeltz[1,2]*

[1]Howard Hughes Medical Institute, Chevy Chase, United States; [2]Department of Molecular, Cellular and Developmental Biology, University of Colorado, Boulder, United States

**Abstract** Mitochondria are dynamic organelles that undergo cycles of fission and fusion at a unified platform defined by endoplasmic reticulum (ER)-mitochondria membrane contact sites (MCSs). These MCSs or nodes co-localize fission and fusion machinery. We set out to identify how ER-associated mitochondrial nodes can regulate both fission and fusion machinery assembly. We have used a promiscuous biotin ligase linked to the fusion machinery, Mfn1, and proteomics to identify an ER membrane protein, ABHD16A, as a major regulator of node formation. In the absence of ABHD16A, fission and fusion machineries fail to recruit to ER-associated mitochondrial nodes, and fission and fusion rates are significantly reduced. ABHD16A contains an acyltransferase motif and an α/β hydrolase domain, and point mutations in critical residues of these regions fail to rescue the formation of ER-associated mitochondrial hot spots. These data suggest a mechanism whereby ABHD16A functions by altering phospholipid composition at ER-mitochondria MCSs. Our data present the first example of an ER membrane protein that regulates the recruitment of both fission and fusion machineries to mitochondria.

*For correspondence: gia.voeltz@colorado.edu

**Competing interest:** The authors declare that no competing interests exist.

## Editor's evaluation

The authors have used state-of-the-art tools to discover and visualize the role of a known ER-localized lipid hydrolase/acyl transferase in creating lipids that facilitate the localization of proteins required for mitochondrial fission and fusion at nodal points of interaction between the ER and mitochondria. The data are clear, quantitative, and compelling with respect to the role of this protein in the processes of mitochondrial constriction, fission, and fusion.

## Introduction

Cells maintain a characteristic mitochondrial architecture important for cellular metabolism and function. Mitochondria maintain their overall architecture and morphology by undergoing cycles of fission and fusion (*Friedman et al., 2010*; *Twig et al., 2008*; *Youle and van der Bliek, 2012*). Disruption of these cycles results in fragmentation or elongation, which can be detrimental to cell health and is associated with various disease states (*Rambold et al., 2011*; *Wai and Langer, 2016*). Both processes are first initiated by the endoplasmic reticulum (ER) at ER-mitochondria membrane contact sites (MCSs). Several factors have been linked to ER-associated mitochondrial fission, including two actin nucleators, INF2 and Spire1c, which are proposed to polymerize actin at ER-mitochondria MCSs to initiate constriction of the outer mitochondrial membrane (OMM; *Korobova et al., 2013*; *Manor et al., 2015*). Subsequently, two GTPases (Drp1 and Dyn2) are recruited to the OMM at ER MCSs to further constrict the OMM, which leads to mitochondrial division (*Bleazard et al., 1999*; *Ferguson*

*and De Camilli, 2012*; *Labrousse et al., 1999*; *Lee et al., 2016*; *Smirnova et al., 1998*). ER-mitochondria MCSs also dictate sites where OMM fusion occurs via Mfn1/2 oligomerization in trans to drive membrane fusion upon GTP hydrolysis (*Abrisch et al., 2020*; *Chen et al., 2003*; *Guo et al., 2018*; *Santel and Fuller, 2001*). Subsequently, inner mitochondrial membrane (IMM) fusion occurs via the GTPase, Opa1 (*Ban et al., 2017*; *Herlan et al., 2003*; *Lee et al., 2004*; *Legros et al., 2002*; *Meeusen et al., 2006*; *Misaka et al., 2002*). However, how the ER contributes to defining a site on mitochondria that is primed for constriction and sufficient to coordinate the recruitment of both fission and fusion machineries and further how these two seemingly opposing activities are co-recruited and also balanced is unclear.

We have recently discovered that cycles of fission and fusion occur at the same location or hot spots that are spatially dictated by the ER. These predefined branch points, or nodes, are where both fission (Drp1) and fusion (Mfn1) machineries converge (*Abrisch et al., 2020*). However, it is not known how these nodes are formed or regulated. We hypothesized that an ER membrane protein facilitates the formation of these nodes. To identify an ER machinery involved in node formation, we have taken advantage of a promiscuous biotin ligase (TurboID) that can biotinylate proteins within a ~10–30 nm range upon biotin addition (*Branon et al., 2018*; *Roux et al., 2012*). We have fused TurboID to the known fusion machinery, Mfn1, to biotinylate and subsequently identify neighboring ER proteins that could regulate these nodes. Using this strategy, we have identified an ER membrane-localized lipid hydrolase, ABHD16A, that could alter lipid membranes for mitochondrial constriction after ER contact sites are established. ABHD16A is not only required for ER-associated mitochondrial constriction, but it is also the first ER protein to be shown to be required for both fission and fusion machinery to assemble at contact sites.

## Results
### Identification of ER-localized lipid hydrolase, ABHD16A, by proximal proteomics

MCSs between the ER and mitochondria define the position where mitochondria undergo fission and fusion (*Abrisch et al., 2020*; *Friedman et al., 2011*; *Guo et al., 2018*). Surprisingly, the fission and fusion machineries co-localize at the same ER-mitochondria MCSs, not separate ones (*Abrisch et al., 2020*). It is not known how the ER functions mechanistically to define constriction sites on mitochondria for both fission and fusion machinery recruitment. Here, we sought to identify ER factors that regulate the assembly of ER-associated nodes where both mitochondrial fission and fusion machinery are recruited. Our strategy was to fuse TurboID, an optimized promiscuous biotin ligase, to the OMM fusion protein, Mfn1 (*Figure 1A*), which assembles at ER-mitochondria MCSs (*Abrisch et al., 2020*; *Branon et al., 2018*). TurboID can biotinylate proteins within a ~10–30 nm range upon the addition of biotin (*Branon et al., 2018*; *Roux et al., 2012*). By immunofluorescence, V5-TurboID-Mfn1 (magenta, with V5 antibody) co-localized well with GFP-Mfn1 (green) puncta on mitochondria (mito-BFP, blue) in HeLa cells (*Figure 1B*, top panels). As a negative control, we similarly fused TurboID to a GTPase domain mutant Mfn1E209A (*Figure 1A and B*). This mutant is deficient in its ability to hydrolyze GTP, does not homodimerize, cannot drive fusion (*Cao et al., 2017*; *Sloat et al., 2019*), and does not enrich at ER-mitochondria MCSs with wild type (WT) GFP-Mfn1 puncta (compare magenta to green, *Figure 1B*, bottom panels) consistent with previous reports (*Abrisch et al., 2020*).

TurboID proximity biotinylation experiments were performed by transfecting HeLa cells with low levels of either V5-TurboID-Mfn1 or V5-TurboID-Mfn1E209A followed by treatment with 500 µM biotin for 3 hr (*Figure 1C*). 3 hr was the minimal time frame to produce robust biotinylation profiles upon expression of low levels of each construct (*Figure 1F*). After biotinylation, the ER membrane fraction was enriched by differential centrifugation as described previously (*Hoyer et al., 2018*; *Wieckowski et al., 2009*). Light membranes were pelleted at 20,000× g and immunoblot analysis was used to confirm that both fractions were enriched with ER and mostly depleted of mitochondria and cytosol (*Figure 1D and E*). The biotinylation profile of the light membrane fraction with a streptavidin-horseradish peroxidase (HRP) probe confirmed that both fusion constructs were enzymatically active (*Figure 1F*). The resulting biotinylated light membrane fraction was purified on streptavidin columns and analyzed by mass spectrometry. Similar self-biotinylation levels of Mfn1 WT vs. mutant were seen in each condition via LFQ (label free quantitation) intensity (*Figure 1G*). The ER membrane fraction

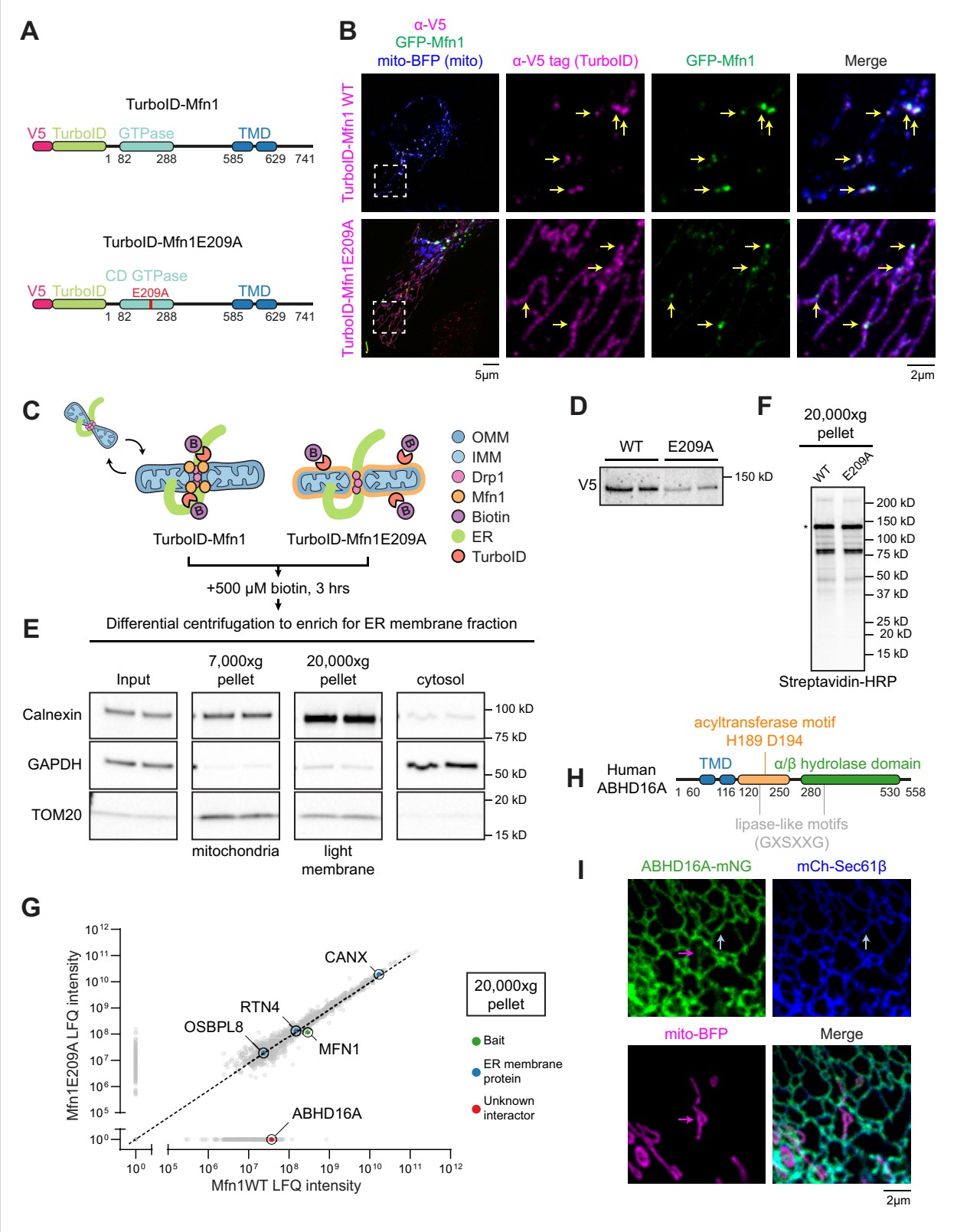

**Figure 1.** Identification of endoplasmic reticulum (ER)-localized lipid hydrolase, ABHD16A, by proximal proteomics. (**A**) Cartoon diagram and domain organization of V5-TurboID-Mfn1 and V5-TurboID-Mfn1E209A (human Mfn1 and indicated amino acid numbers). V5-tagged TurboID was added to the N-terminus of Mfn1. Red E209A indicates the catalytically dead (CD) mutation created in the GTPase domain. TMD: transmembrane domain. (**B**) Representative images and insets of a HeLa cell expressing GFP-Mfn1 (green), mito-BFP (blue), and immunostained with antibody against the V5

*Figure 1 continued on next page*

*Figure 1 continued*

tag (magenta, to detect Mfn1 constructs). Yellow arrows indicate GFP-Mfn1 puncta along mitochondria. (**C**) Cartoon diagram of the strategy used in HeLa cells to biotinylate ER-mitochondria membrane contact site (MCS) proteins with V5-TurboID-Mfn1 vs. V5-TurboID-Mfn1E209A. OMM: outer mitochondrial membrane (blue); IMM: inner mitochondrial membrane (light blue); Drp1: Dynamin-related protein 1 (fission machinery, pink); Mfn1: Mitofusin 1 (fusion machinery, orange); B: biotin. (**D**) Immunoblot analysis (anti-V5) shows relative expression of V5-TurboID-Mfn1 and V5-TurboID-Mfn1E209A in HeLa cells. (**E**) Immunoblot analyses of fractions collected by differential centrifugation including: 7000× g pellet containing mitochondria (TOM20, mitochondria), 20,000× g pellet containing light membrane and ER (Calnexin, ER), and supernatant containing cytosol (GAPDH, cytosol). (**F**) The 20,000× g pellet was solubilized and probed with streptavidin HRP to reveal biotinylation profiles for each sample prior to mass spectrometry. Asterisk denotes a band size indicative of construct self-biotinylation. (**G**) Average LFQ intensities of proteins biotinylated and enriched in Mfn1 wild type (WT) vs. Mfn1E209A sample in the 20,000× g pellet. Dashed line indicates equivalent enrichment in the WT vs. E209A mutant sample. Data from LFQ intensity are the average of two technical replicates. CANX: Calnexin; RTN4: Reticulon-4; MFN1: Mitofusin-1; OSBPL8: Oxysterol binding protein like 8; ABHD16A: Alpha/beta hydrolase domain containing phospholipase 16 A. (**H**) Cartoon diagram of human ABHD16A with motif or domain annotations. (**I**) Representative inset of a U-2 OS cell expressing low levels of ABHD16A-mNG (green), mCh-Sec61β (ER, blue), and mito-BFP (mitochondria, magenta). Arrows indicate ABHD16A localization to ER (blue) and mitochondria (magenta). Scale bar = 5 μm or 2 μm for insets. See *Figure 1—source data 1*, *Figure 1—source data 2*, *Figure 1—source data 3*, *Figure 1—source data 4*, *Figure 1—source data 5*, *Figure 1—source data 6*, *Figure 1—source data 7*, *Figure 1—source data 8*, *Figure 1—source data 9*, *Figure 1—source data 10*.

The online version of this article includes the following source data and figure supplement(s) for figure 1:

**Source data 1.** Related To *Figure 1D*.

**Source data 2.** Related to *Figure 1E*.

**Source data 3.** Related to *Figure 1F*.

**Source data 4.** Related to *Figure 1G*.

**Source data 5.** Related to *Figure 1D*.

**Source data 6.** Related to *Figure 1D*.

**Source data 7.** Related to *Figure 1E*.

**Source data 8.** Related to *Figure 1E*.

**Source data 9.** Related to *Figure 1F*.

**Source data 10.** Related to *Figure 1F*.

**Figure supplement 1.** Endogenous ABHD16A localizes to the endoplasmic reticulum (ER) and mitochondria, to a lesser extent (related to *Figure 1*).

**Figure supplement 1—source data 1.** Related to *Figure 1—figure supplement 1B*.

**Figure supplement 1—source data 2.** Related to *Figure 1—figure supplement 1B*.

**Figure supplement 1—source data 3.** Related to *Figure 1—figure supplement 1B*.

**Figure supplement 1—source data 4.** Related to *Figure 1—figure supplement 1B*.

**Figure supplement 1—source data 5.** Related to *Figure 1—figure supplement 1B*.

**Figure supplement 1—source data 6.** Related to *Figure 1—figure supplement 1B*.

---

was equivalently enriched in both conditions because known ER membrane proteins, such as calnexin (CANX), Reticulon-4 (RTN4), and Oxysterol binding protein like 8 (OSBPL8/ORP8), were identified to equal levels (*Figure 1G*). Upon further investigation of the top protein candidates, we identified an ER membrane protein, ABHD16A, that was highly enriched in the WT Mfn1 sample (19th highest enrichment) as a candidate effector of ER-associated mitochondrial node formation (*Figure 1G* and *Figure 1—figure supplement 1C*). The top 18 proteins, prior to ABHD16A, were either not localized to the ER or in the case of ARL6IP1, were characterized as an ER-shaping protein (*Figure 1—figure supplement 1C* and *Yamamoto et al., 2014*).

ABHD16A is reported to be a phospholipid hydrolase, with the highest affinity for phosphotidylserine (PS) lipolysis that resides on the ER membrane via two transmembrane domains (TMDs; *Kamat et al., 2015*; *Lord et al., 2013*; *Singh et al., 2020*). It is highly conserved in vertebrates (96% conserved from human to mouse) and contains an alpha/beta hydrolase domain responsible for its phospholipid hydrolase activity, a predicted acyltransferase motif (H189/D194), and two predicted lipase-like motifs (GXSXXG with serines at positions S176 and S306), all of which face the cytoplasm (*Figure 1H*; *Kamat et al., 2015*; *Lord et al., 2013*; *Singh et al., 2020*). Phospholipid hydrolases are known to remove one acyl chain from phospholipids, whereas acyltransferase motifs can catalyze the opposite reaction of a phospholipid hydrolase: where a single-chain phospholipid reacts with an acyl-CoA (coenzyme A) to restore a dual-chain phospholipid (*Aguado and Campbell, 1998*;

*Eberhardt et al., 1997*; *Lord et al., 2013*; *Zhao et al., 2008*). The opposing predicted acyltransferase motif in ABHD16A, although biochemically uncharacterized, is present in many integral ER membrane proteins and other ABHD family proteins and has the potential to either add or alter acyl chains to lysophospholipids/phospholipids (*Bononi et al., 2021*; *Harayama et al., 2014*; *Hishikawa et al., 2014*; *Shindou et al., 2013*; *Zhao et al., 2008*). Lipase-like motifs (GXSXXG) are predicted to have similarity to bacterial lipase motifs (GXSXG), although it is unknown whether ABHD16A's lipase-like motifs, one near the acyltransferase motif, and one within the alpha/beta hydrolase domain, have lipase activity.

We generated a fluorescently tagged ABHD16A (ABHD16A-mNeonGreen) to determine its localization in U-2 OS cells by live confocal fluorescence microscopy. U-2 OS cells were co-transfected with low levels of ABHD16A-mNeonGreen (green), a mitochondrial matrix marker (mito-BFP, magenta), and an ER membrane marker (mCherry-Sec61β, blue). At low expression levels, ABHD16A-mNG co-localized homogenously along the ER membrane (blue arrow) and to a much lesser degree on mitochondria (magenta arrow) but did not appear to accumulate at MCSs (*Figure 1I* and *Figure 1—figure supplement 1A*). Endogenous ABHD16A localization was also analyzed by immunoblot of pure, crude, and mitochondrial-associated membrane (MAM) fractions isolated from U-2 OS cells. These data showed that endogenous ABHD16A can be found in the MAM similar to other ER proteins and to a lesser extent in the pure mitochondrial fraction, as previously reported (*Figure 1—figure supplement 1B*; *Singh et al., 2020*; *Wieckowski et al., 2009*).

## ABHD16A is required for the formation of ER-associated fission and fusion hot spots

ABHD16A was preferentially biotinylated by TurboID-Mfn1 WT. Therefore, we assayed whether ABHD16A depletion affects the recruitment of the OMM fusion machinery, Mfn1, to ER-mitochondria MCSs. U-2 OS cells were co-transfected with fluorescently tagged Mfn1 (GFP-Mfn1, green), a mitochondrial matrix marker (mito-BFP, magenta), and an ER marker (mNG-Sec61β, blue) and with either control (siCTRL) or ABHD16A siRNA. Immunoblot analysis reveals that ABHD16A can be efficiently depleted by siRNA transfected into U-2 OS cells (*Figure 2—figure supplement 1D*). Mfn1 recruitment efficiency was scored as the number of Mfn1 puncta present per micron of mitochondrial length. In siCTRL-treated cells, GFP-Mfn1 accumulates at puncta along mitochondria (*Figure 2A*, top panel), as expected (*Abrisch et al., 2020*). ABHD16A depletion significantly reduced Mfn1 puncta compared to siCTRL-treated cells (*Figure 2A and D*: ~0.1 vs. 0.3 puncta per μm of mitochondria, respectively). Mfn1 puncta were restored by re-expression of an siRNA-resistant ABHD16A-Halo construct (*Figure 2A and D*, and *Figure 2—figure supplement 1A*). These data demonstrate that ABHD16A regulates localization of Mfn1 to ER-mitochondria contact sites.

Since fission machinery has been shown to co-localize with fusion machinery, we tested whether ABHD16A depletion similarly disrupts the recruitment of fission machinery, Drp1, to ER-mitochondria MCSs. U-2 OS cells were co-transfected with fluorescently tagged Drp1 (mCh-Drp1, green), mito-BFP (magenta), and mNG-Sec61β (ER, blue) and with either siCTRL or ABHD16A siRNA (*Figure 2B* and *Figure 2—figure supplement 1B*). The efficiency of fission machinery recruitment was scored as the percent of ER-mitochondria crossings that co-localize with a Drp1 puncta. Drp1 recruitment to crossings was significantly reduced in ABHD16A-depleted vs. siCTRL-treated cells (18% vs. 47%, respectively, *Figure 2E*). The recruitment of Drp1 to ER-associated puncta could be restored by reintroduction of an siRNA-resistant ABHD16A expression construct (ABHD16A-Halo; *Figure 2B and E*, and *Figure 2—figure supplement 1B*). These data reveal that ABHD16A is also required for Drp1 fission machinery recruitment to ER-mitochondria MCSs.

Next, we measured whether ABHD16A is required for the assembly of ER-associated mitochondrial nodes, which are locations where Drp1 and Mfn1 machineries co-localize (*Abrisch et al., 2020*). Cells were co-transfected with mCh-Drp1 (magenta), mNG-Mfn1 (green), and mito-BFP (not shown) and with either siCTRL or ABHD16A siRNA and imaged live (*Figure 2C* and *Figure 2—figure supplement 1C*). Node density was scored as the number of puncta containing both Drp1 and Mfn1 per μm of mitochondrial length. ABHD16A-depleted cells had significantly fewer nodes than control cells (~0.13 vs.~0.34 puncta per μm of mitochondria, *Figure 2F*). Mitochondrial node formation could be rescued by re-expression of siRNA-resistant ABHD16A-Halo but not the empty vector (EV) control (*Figure 2C and F*, and *Figure 2—figure supplement 1C*). Together, these data demonstrate that ABHD16A is

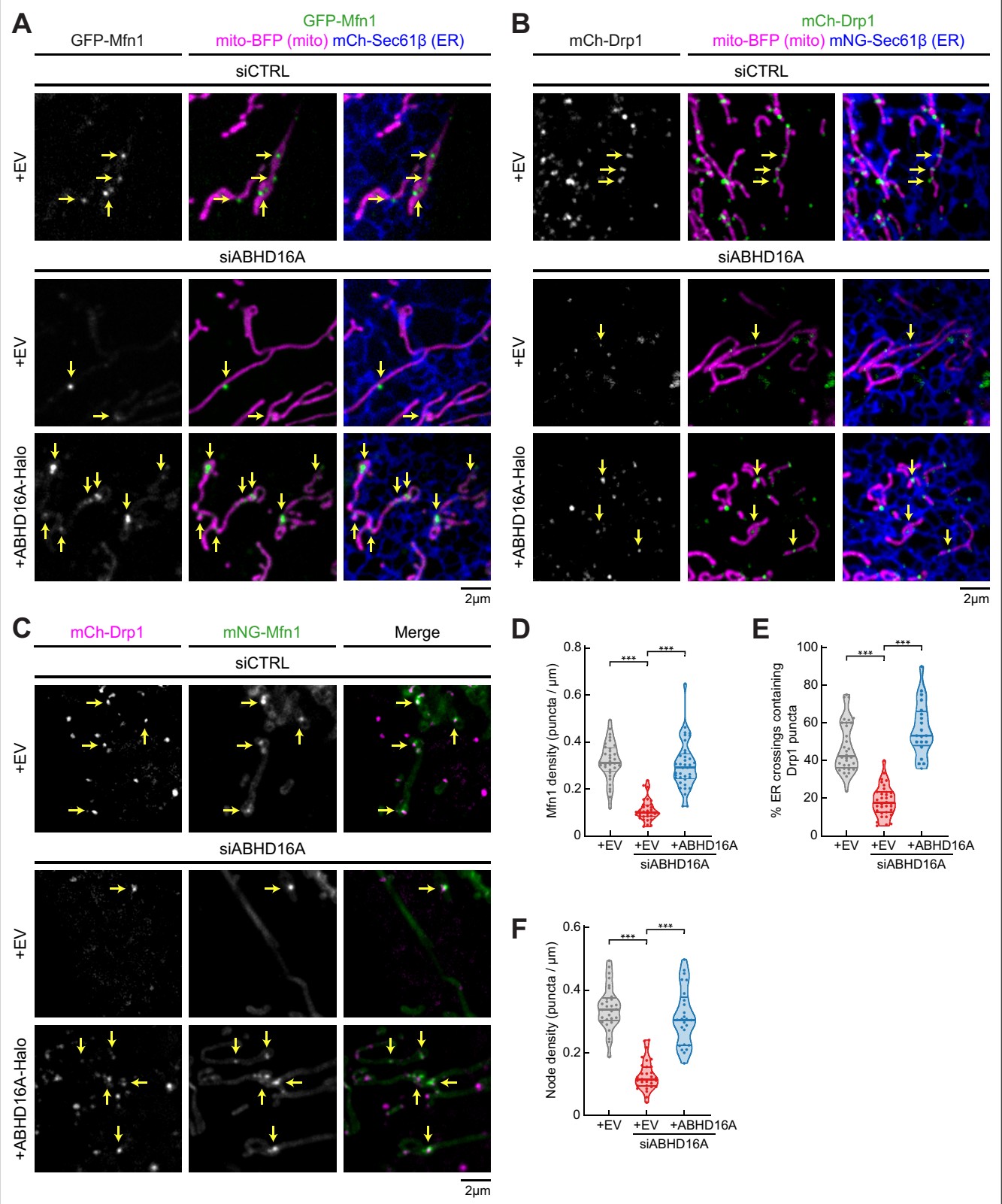

**Figure 2.** ABHD16A is required for the formation of endoplasmic reticulum (ER)-associated fission and fusion hot spots. (**A**) Representative images of U-2 OS cells transfected with GFP-Mfn1 (green), mito-BFP (magenta), mCh-Sec61β (ER, blue), and either control siRNA (n=33 cells, top), ABHD16A siRNA (n=33 cells, middle), or ABHD16A siRNA rescued with ABHD16A-Halo (n=37 cells, bottom). Yellow arrows indicate examples of Mfn1 puncta along mitochondria at ER-mitochondria crossings. (**B**) Representative images of U-2 OS cells transfected with mCh-Drp1 (green), mito-BFP (magenta),

*Figure 2 continued on next page*

*Figure 2 continued*

mNG-Sec61β (ER, blue), and either control siRNA (n=32 cells, top), ABHD16A siRNA (n=34 cells, middle), or ABHD16A siRNA rescued with ABHD16A-Halo (n=20 cells, bottom). Yellow arrows indicate examples of Drp1 puncta at ER-mitochondria crossings. (**C**) Representative images of U-2 OS cells transfected with mCh-Drp1 (magenta), GFP-Mfn1 (green), mito-BFP (not shown), and either control siRNA (n=29 cells, top), ABHD16A siRNA (n=31 cells, middle), or ABHD16A siRNA rescued with ABHD16A-Halo (n=25 cells, bottom). Yellow arrows indicate examples of nodes along mitochondria. (**D**) Quantification of Mfn1 density along mitochondrial length from experiments shown in (**A**). (**E**) Quantification of percent ER crossings containing Drp1 puncta from experiments shown in (**B**). (**F**) Quantification of node density along mitochondrial length from experiments shown in (**C**). All data were taken from three biological replicates; statistical significance was calculated by one-way ANOVA. ***p≤0.001. Scale bar = 2 µm. See *Figure 2—source data 1*.

The online version of this article includes the following source data and figure supplement(s) for figure 2:

**Source data 1.** Related to *Figure 2D–F*.

**Figure supplement 1.** ABHD16A depletion decreases fusion/fission node formation (related to *Figure 2*).

**Figure supplement 1—source data 1.** Related to *Figure 2—figure supplement 1D*.

**Figure supplement 1—source data 2.** Related to *Figure 2—figure supplement 1D*.

**Figure supplement 1—source data 3.** Related to *Figure 2—figure supplement 1D*.

required for the co-localization of fission and fusion machineries to mitochondrial nodes, suggesting a common mechanism is required for their recruitment.

## ABHD16A is required for efficient cycles of ER-mediated mitochondrial fission and fusion

We predicted that ABHD16A depletion would also reduce mitochondrial fission and fusion rates concomitant with its deleterious effect on fission and fusion machinery recruitment to ER-associated nodes. To test this directly, we first scored fusion rates by using a photoconvertible fluorophore to label the OMM mMAPLE-OMP25, as previously described (*Abrisch et al., 2020*). Briefly, upon 405 nm laser stimulation, mMAPLE fluoresces red instead of green and fusion can be scored by red fluorescence diffusion (*Figure 3—figure supplement 1A*). Cells were co-transfected with mMAPLE-OMP25 and either siCTRL or ABHD16A siRNA, single mitochondria were photoconverted from green to red (magenta in panels), and fusion rates were scored visually and quantitatively during live 5 min time-lapse movies by the observation of fluorescence mixing between two mitochondria (*Figure 3A–D*, *Figure 3—figure supplement 1B*, *Figure 3—video 1*, *Figure 3—video 2*, *Figure 3—video 3*). Fusion rates were significantly reduced in ABHD16A-depleted cells (*Figure 3*, *Figure 3—figure supplement 1B*, and *Figure 3—video 1*, *Figure 3—video 2* from a rate of 0.25 [siCTRL] to 0.06 [siABHD16A] fusion events per mitochondrion/minute). Fusion rates could be rescued by reintroduction of siRNA-resistant ABHD16A-Halo to depleted cells (*Figure 3*, *Figure 3—figure supplement 1B*, and *Figure 3—video 3*).

To further investigate the mechanism by which ABHD16A depletion disrupts mitochondrial fusion, we binned mitochondrial fusion events into two categories: tip-to-middle fusion and tip-to-tip fusion. Tip-to-middle fusion occurs when one mitochondrial tip marked with Mfn1 travels to an Mfn1-marked spot along the middle of a mitochondrion. Tip-to-tip fusion occurs when two mitochondrial tips marked with Mfn1 come together to fuse. In siCTRL cells, the majority of events occur by tip-to-middle fusion (63 out of 85 fusion events, 74% of total fusion events, 0.19 out of 0.25 fusion events per mitochondrion/min, *Figure 3F*). Upon ABHD16A depletion, tip-to-middle fusion events are highly reduced (0.19 in siCTRL vs. 0.02 tip-to-middle fusion events per mitochondrion/min in ABHD16A depletion), whereas tip-to-tip fusion is less affected (0.06 in siCTRL vs. 0.04 tip-to-tip fusion events per mitochondrion/min in ABHD16A depletion, *Figure 3F*). What makes these results notable is that the 'middle' part of the tip-to-middle fusion event occurs at ER-associated mitochondrial constriction sites where fission and fusion machineries also assemble and co-localize (*Abrisch et al., 2020*).

Next, we scored the effect of ABHD16A depletion on the rate of fission (within the same experiment that was used to score fusion rates). Indeed, fission rates were also significantly reduced upon ABHD16A depletion (a > threefold reduction from 0.17 to 0.05 fission events per mitochondrion/min; *Figure 3E* and *Figure 3—figure supplement 1C*). Fission rates could also be restored by expressing exogenous siRNA-resistant ABHD16A-Halo and not with an EV control (Halo-N1; *Figure 3E* and *Figure 3—figure supplement 1C*). These data show that ABHD16A depletion reduces fusion and fission rates concomitant with the reduced recruitment of fusion and fission machineries to

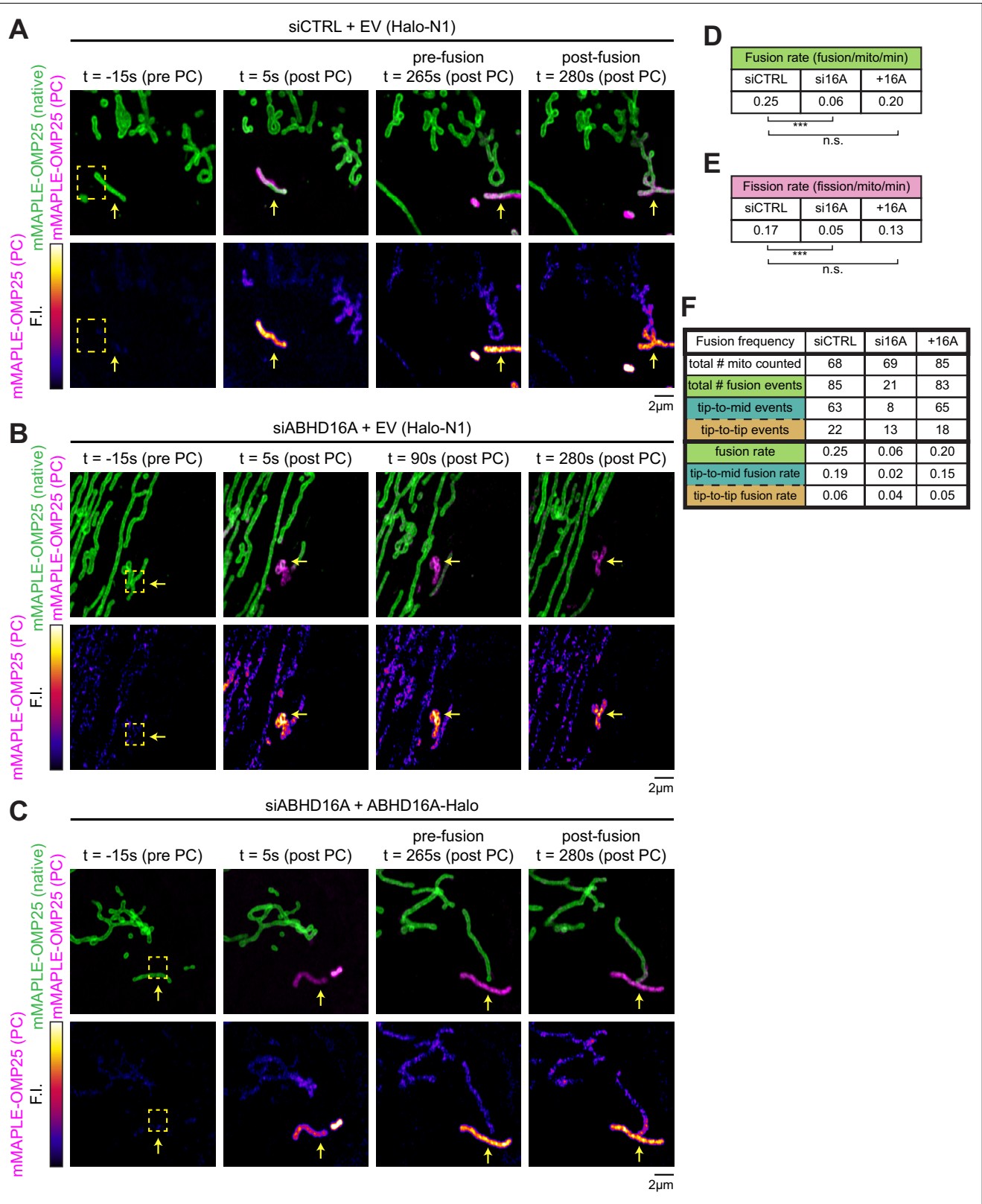

**Figure 3.** ABHD16A is required for efficient cycles of endoplasmic reticulum (ER)-mediated mitochondrial fission and fusion. (**A**) Representative time lapse of a live U-2 OS cell over a 5 min movie expressing mMAPLE-OMP25 and control siRNA (n=30 cells). Top panel: green shows native mMAPLE-OMP25 signal, while magenta shows photoconverted (PC) mMAPLE-OMP25. Bottom panel: fire lookup table (LUT) of PC mMAPLE-OMP25 to show fusion event. Yellow box indicates region of interest (ROI) exposed to 405 nm light. Yellow arrow indicates photoconverted mitochondrion. (**B**) As

*Figure 3 continued on next page*

*Figure 3 continued*

in (**A**) for live U-2 OS cell expressing mMAPLE-OMP25 and ABHD16A siRNA (n=29 cells). Fire LUT and yellow arrow show no fusion occurring over the 5 min movie. (**C**) As in (**A**) for live U-2 OS cell expressing mMAPLE-OMP25, ABHD16A siRNA, and rescued with ABHD16A-Halo (n=31 cells). (**D**) Quantification of fusion rate per mitochondrion per minute from experiments shown in (**A–C**). (**E**) Quantification of fission rate per mitochondrion per minute from experiments shown in (**A–C**). (**F**) Quantification of the two types of fusion events per mitochondrion per minute from experiments shown in (**A–D**). Table displays total number of mitochondria, total fusion events, types of fusion events, and rates of fusion events (fusion/mitochondrion/minute). All data were taken from three biological replicates; statistical significance was calculated by one-way ANOVA. n.s., not significant; ***p≤0.001. Scale bar = 2 μm. See *Figure 3—source data 1* and *Figure 3—video 1*, *Figure 3—video 2*, *Figure 3—video 3*.

The online version of this article includes the following video, source data, and figure supplement(s) for figure 3:

**Source data 1.** Related to *Figure 3D–F*.

**Figure supplement 1.** ABHD16A depletion reduces fusion and fission rate (related to *Figure 3*).

**Figure 3—video 1.** siCTRL-treated U-2 OS cell labeled with photoconvertible mMAPLE-OMP25 from Figure 3A.
https://elifesciences.org/articles/84279/figures#fig3video1

**Figure 3—video 2.** ABHD16A siRNA treated U-2 OS cell labeled with photoconvertible mMAPLE-OMP25 from Figure 3B.
https://elifesciences.org/articles/84279/figures#fig3video2

**Figure 3—video 3.** ABHD16A siRNA treated U-2 OS cell co-transfected with wild type (WT) ABHD16A-Halo and mMAPLE-OMP25 from Figure 3C.
https://elifesciences.org/articles/84279/figures#fig3video3

ER-associated nodes. Taken together, ABHD16A is a regulator of ER-associated mitochondrial fission and fusion dynamics.

## ER-localized ABHD16A is required to maintain mitochondrial morphology

Since ABHD16A depletion alters fission and fusion machinery recruitment, we scored the overall effect of ABHD16A depletion on mitochondrial morphology as previously described (*Lee et al., 2016*). U2-OS cells were co-transfected with mito-BFP and either siCTRL or ABHD16A siRNA to deplete ABHD16A. On average, ABHD16A-depleted cells had highly elongated mitochondria compared to control (4.4 μm² vs. 2.2 μm², respectively, *Figure 4B, D and E*, and *Figure 4—figure supplement 1A*). Morphology could be restored to siCTRL lengths by re-expressing siRNA-resistant ABHD16A (ABHD16A-mNG; *Figure 4C and D*; also see immunoblot in *Figure 4E* for relative expression levels). The depletion of ABHD16A also caused mitochondrial elongation (compared to control) in HeLa cells, showing that this effect is not cell-type specific (*Figure 4—figure supplement 1C–1E*). Consistently, ABHD16A KO U-2 OS cells also display an elongated mitochondrial morphology and reduced Drp1 recruitment (*Figure 4—figure supplement 1F* and *Figure 4—figure supplement 1G*). Since ABHD16A depletion significantly impairs tip-to-middle fusion and overall fission rates, we reasoned that the residual tip-to-tip mitochondrial fusion (which can still occur upon depletion) is sufficient to promote the elongated mitochondrial phenotype.

Our localization experiments revealed that although ABHD16A predominantly localizes to the ER, a small amount of ABHD16A can also be seen on mitochondria (*Figure 1I*, *Figure 1—figure supplement 1A*, and *Figure 1—figure supplement 1B*). To understand how ABHD16A functions mechanistically, it is thus important to understand whether the ER-localized or the trace mitochondrial-localized ABHD16A pool is responsible for regulating mitochondrial dynamics. We therefore tested whether expression of an exclusively ER-targeted or mitochondrial-targeted ABHD16A functions to rescue mitochondrial morphology. To target ABHD16A exclusively to the ER or to mitochondria, we replaced the N-terminal TMDs of ABHD16A with either the N-terminal TMD of the ER-localized protein Stim1 or the N-terminal TMD of the mitochondrial-localized protein Tom70 (*Figure 4A*). Both chimeric constructs localize as expected exclusively to the ER or mitochondria, respectively (*Figure 4—figure supplement 1B*). However, only re-expression of the siRNA-resistant ER-localized ABHD16A (ABHD16A^ER) rescues mitochondrial morphology, while rescue with the mitochondria-localized version (ABHD16A^mito) does not (*Figure 4C-E*). In a complementary experiment, we also scored the effect of ABHD16A overexpression (OE) on mitochondrial morphology. On average, ABHD16A OE cells had a 3.6-fold reduction in area per mitochondrion (*Figure 4F and G*). Consistent with rescue data above (*Figure 4B–E*, *Figure 4—figure supplement 1A*, and *Figure 4—figure supplement 1B*), only the ER-localized ABHD16A (ABHD16A^ER) drove mitochondrial fragmentation, whereas the

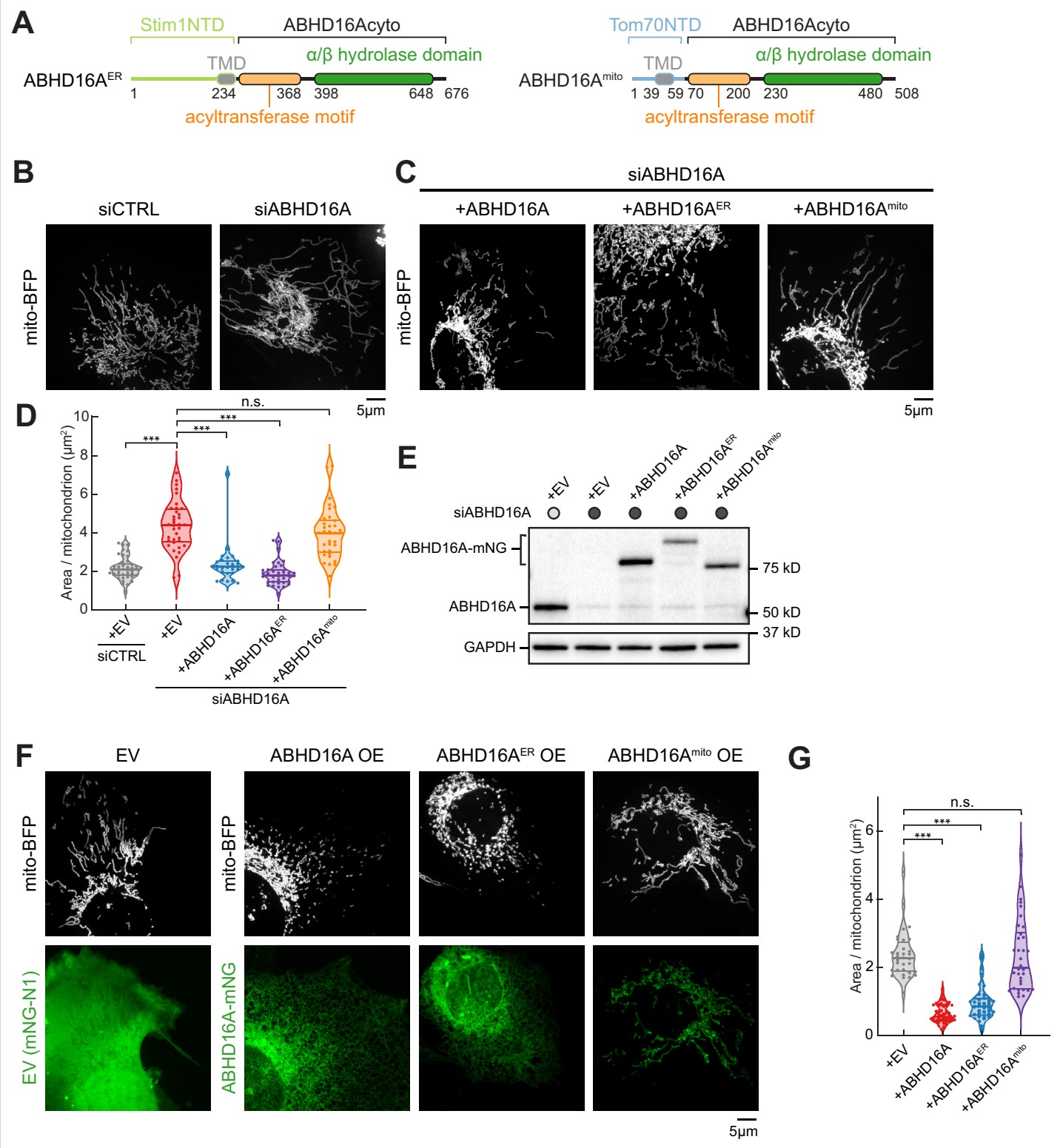

**Figure 4.** Endoplasmic reticulum (ER)-localized ABHD16A is required to maintain mitochondrial morphology. (**A**) Domain organization of chimeric constructs for ER-localized ABHD16A (ABHD16A^ER) via Stim1's N-terminal domain (NTD; left) or mitochondrial-localized ABHD16A (ABHD16A^mito) via Tom70's NTD (right). TMD: transmembrane domain. (**B**) Representative images of mitochondrial morphology (labeled by mito-BFP, gray) of U-2 OS cells transfected with control siRNA (siCTRL, n=41 cells, left) or ABHD16A siRNA (siABHD16A, n=36 cells, right). (**C**) Representative images of mitochondrial morphology (labeled by mito-BFP, gray) of U-2 OS cells transfected with ABHD16A siRNA and rescued with either siRNA-resistant ABHD16A-mNG

*Figure 4 continued on next page*

*Figure 4 continued*

(n=28 cells, left), ABHD16A^ER-mNG (n=36 cells, middle), or ABHD16A^mito-mNG (n=34 cells, right). (**D**) Quantification of mean mitochondrial size (area per mitochondrion in μm²) within a 15×15 μm region of interest (ROI) from (**B**) and (**C**). (**E**) Immunoblot shows efficiency of depletion in cells treated with control siRNA or ABHD16A siRNA and rescued with siRNA-resistant ABHD16A-mNG constructs. GAPDH serves as a loading control. (**F**) Representative images of mitochondrial morphology of U-2 OS cells (labeled by mito-BFP, gray) and either empty vector (EV, control, n=32 cells), ABHD16A-mNG (n=40 cells), ABHD16A^ER-mNG (n=38 cells), or ABHD16A^mito-mNG (n=34 cells) in green. (**G**) Quantification of mean mitochondrial size (area per mitochondrion in μm²) within a 15×15 μm ROI from (**F**). All data were taken from three biological replicates; statistical significance was calculated by one-way ANOVA. n.s., not significant; ***p≤0.001. Scale bar = 5 μm. See *Figure 4—source data 1*, *Figure 4—source data 2*, *Figure 4—source data 3*, *Figure 4—source data 4*.

The online version of this article includes the following source data and figure supplement(s) for figure 4:

**Source data 1.** Related to *Figure 4D and G*.

**Source data 2.** Related to *Figure 4E*.

**Source data 3.** Related to *Figure 4E*.

**Source data 4.** Related to *Figure 4E*.

**Figure supplement 1.** ABHD16A is required to maintain mitochondrial morphology in HeLa cells (related to *Figure 4*).

**Figure supplement 1—source data 1.** Related to *Figure 4—figure supplement 1D*.

**Figure supplement 1—source data 2.** Related to *Figure 4—figure supplement 1E*.

**Figure supplement 1—source data 3.** Related to *Figure 4—figure supplement 1D*.

**Figure supplement 1—source data 4.** Related to *Figure 4—figure supplement 1D*.

**Figure supplement 1—source data 5.** Related to *Figure 4—figure supplement 1G*.

**Figure supplement 1—source data 6.** Related to *Figure 4—figure supplement 1G*.

**Figure supplement 1—source data 7.** Related to *Figure 4—figure supplement 1G*.

mitochondrial-localized form (ABHD16A^mito) did not (*Figure 4F and G*). Together, the depletion and OE experiments suggest that the ER-localized ABHD16A regulates mitochondrial morphology from the ER.

## ABHD16A is not a tether

Our data demonstrate that ABHD16A is required for fission and fusion machinery recruitment to ER-mitochondria MCSs. We tested whether this is because ABHD16A is required for tethering between the ER and mitochondria. ABHD16A was depleted by siRNA, and a dimerization-dependent fluorescent protein (ddFP) reporter system was used to score ER-mitochondria MCS, as previously described (*Abrisch et al., 2020*). This reporter system consists of a heterodimeric fluorescent protein complex (dimly red fluorescent protein copy A, RA, and protein binding partner, B; *Ding et al., 2015*). Upon dimerization of RA with B, RA fluorescence increases leading to a detectable red signal. The low binding affinity of the RA/B interaction ($K_d$=~7 μM) ensures that the interaction is reversible and does not cause artificial wrapping, which allows for bona-fide MCSs to be visualized. RA was fused to an ER membrane protein, Sec61β, to target half of the dimer to the ER, and B was fused to the OMM protein Mff, as previously described (*Abrisch et al., 2020*). When the ER and mitochondria come into close proximity (tethering distance), an increased fluorescence intensity is observed (*Figure 5A*). Cells were co-transfected with either control or ABHD16A siRNA, mito-BFP (magenta), an ER marker (mNG-KDEL, blue), ddFP reporters (RA-Sec61β and B-Mff, dimer will fluoresce green) and were rescued with siRNA-resistant ABHD16A (ABHD16A-SNAP) or an EV control (SNAP-N1; *Figure 5B*, *Figure 5—figure supplement 1A*, and *Figure 5—figure supplement 1B*). ER-mitochondria contact was quantified by assessing binary ddFP overlap with mitochondria as a proportion of total mitochondrial area within a 20×20 μm² region of interest (ROI). To ensure resolvable regions were quantified in each condition, we also measured the binary ER overlap with mitochondria as a proportion of total mitochondrial area, which was notably similar throughout each condition (*Figure 5C*). Thus, ABHD16A depletion leads to 2.1-fold increase in ER-mitochondria MCS area (ddFP) compared to siCTRL-treated cells (*Figure 5D*). This increased contact was reduced by re-expression of siRNA-resistant ABHD16A (*Figure 5B and D*, *Figure 5—figure supplement 1A*, and *Figure 5—figure supplement 1B*). These data suggest that ABHD16A is not an ER-mitochondria contact site tether.

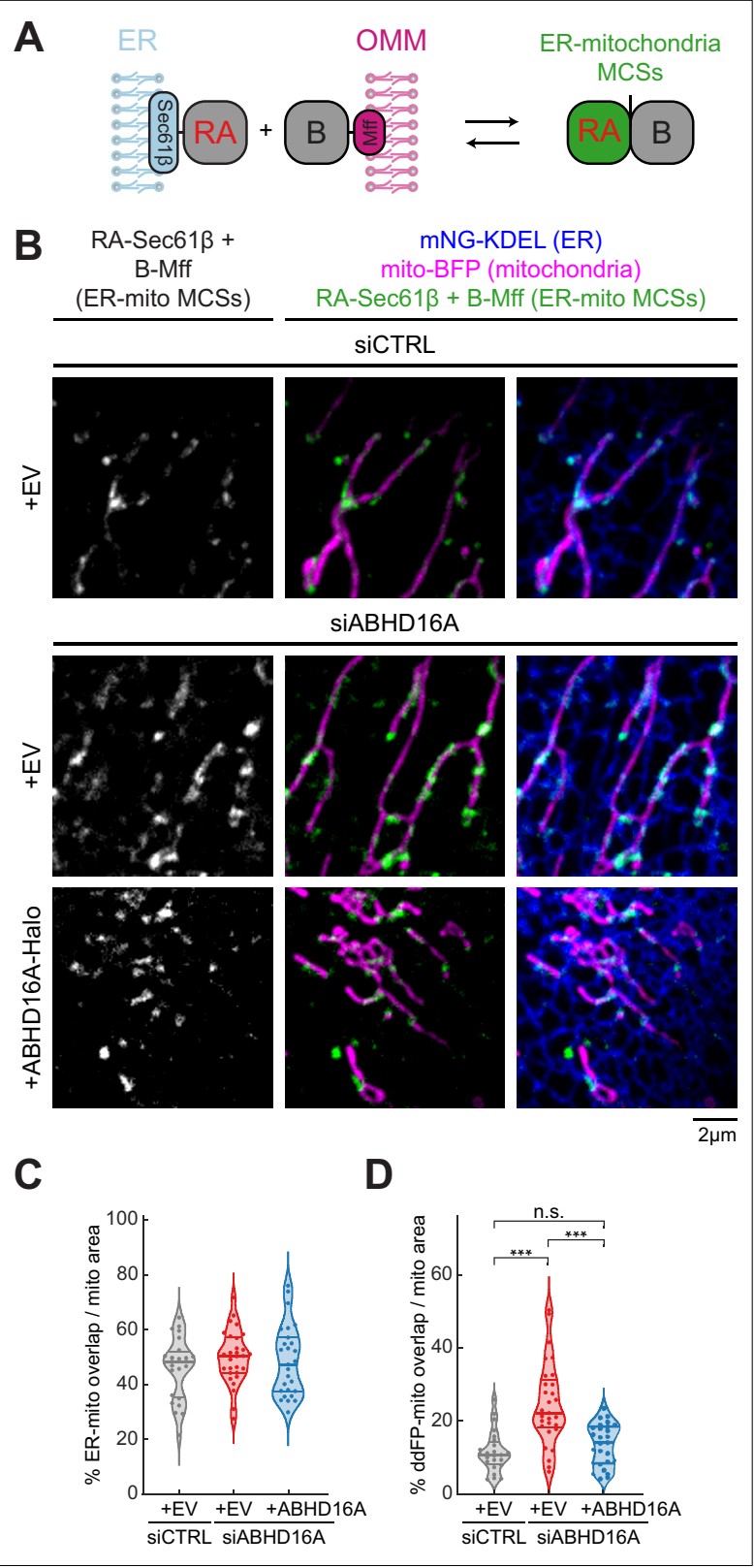

**Figure 5.** ABHD16A is not required for endoplasmic reticulum (ER)-mitochondria membrane contact site (MCS) formation. (**A**) Cartoon depiction of the ER-mitochondria dimerization-dependent fluorescent protein (ddFP) MCS reporter system: one monomer fused to an ER protein (RA-Sec61β) and the other monomer fused to an outer mitochondrial membrane (OMM) protein (B-Mff). When ER and mitochondria come within ~10–30 nm, dimers form

*Figure 5 continued on next page*

*Figure 5 continued*

and fluoresce red (false colored in green) at ER-mitochondria MCSs. (**B**) Representative images of ER-mitochondria MCSs in U-2 OS cells transfected with mNG-KDEL (ER, blue), mito-BFP (magenta), RA-Sec61β and B-Mff (ER-mitochondria ddFP MCS pair, green), and either control siRNA (n=24 cells, top), ABHD16A siRNA (n=29 cells, middle), or ABHD16A siRNA rescued with ABHD16A-Halo (n=27 cells, bottom). (**C**) Quantification of percentage of ER-mitochondria overlap over total mitochondrial area calculated by Mander's correlation coefficient (MCC) of two binary images taken from 20×20 µm region of interests (ROIs) from images in (**B**). (**D**) Quantification of percentage of ddFP-mitochondria overlap over total mitochondrial area calculated by MCC of two binary images taken from 20×20 µm ROIs from images in (**B**). All data were taken from three biological replicates; statistical significance was calculated by one-way ANOVA. n.s., not significant; ***p≤0.001. Scale bar = 2 µm. See *Figure 5—source data 1*.

The online version of this article includes the following source data and figure supplement(s) for figure 5:

**Source data 1.** Related to *Figure 5C and D*.

**Figure supplement 1.** ABHD16A is not required for endoplasmic reticulum (ER)-mitochondria membrane contact site (MCS) formation (related to *Figure 5*).

**Figure supplement 1—source data 1.** Related to *Figure 5—figure supplement 1B*.

**Figure supplement 1—source data 2.** Related to *Figure 5—figure supplement 1B*.

**Figure supplement 1—source data 3.** Related to *Figure 5—figure supplement 1B*.

## ABHD16A is required for ER-associated mitochondrial constriction

We have previously shown that ER-associated fission and fusion occur at ER-associated mitochondrial constriction sites or nodes (*Abrisch et al., 2020*; *Friedman et al., 2011*). Since ABHD16A depletion blocks fission and fusion machinery recruitment, we wondered whether ABHD16A is required for ER-associated mitochondrial constriction. U-2 OS cells were co-transfected with mito-BFP (magenta), an ER marker (mCh-Sec61β, green), and with either ABHD16A siRNA or siCTRL (*Figure 6A*, *Figure 6—figure supplement 1B*, and *Figure 6—figure supplement 1C*). ER-associated mitochondrial constrictions were defined as positions where the fluorescence intensity of the mitochondrial matrix marker (mito-BFP) dipped by >40% along several z-planes (orange arrows; ER crossings with no constriction were marked by a purple arrow, *Figure 6—figure supplement 1A*). ABHD16A-depleted cells had threefold fewer ER-associated mitochondrial constrictions than siCTRL-treated cells (23 vs. 63%, respectively, *Figure 6A and C*, and *Figure 6—figure supplement 1B*). Re-expression of siRNA-resistant ABHD16A-mNG restored levels of ER-associated mitochondrial constrictions to 75% whereas an EV control (mNG-N1) did not (*Figure 6A and C*, and *Figure 6—figure supplement 1B*). These data reveal that ABHD16A is required for ER-associated mitochondrial membrane constriction, which explains why its depletion disrupts fission/fusion machinery recruitment and node formation.

## ABHD16A's alpha/beta hydrolase domain rescues mitochondrial constrictions

A local alteration in phospholipid shape at ER MCSs could change membrane curvature and might be the mechanism used by ER MCSs to constrict mitochondria (*Agrawal and Ramachandran, 2019*; *Harayama and Riezman, 2018*; *van Meer et al., 2008*). Our data have shown that ABHD16A is a determinant of ER-associated mitochondrial constriction. ABHD16A has two compelling and highly conserved motifs/domains predicted to alter lipid shape in opposing ways: an acyltransferase motif and an alpha/beta hydrolase domain consisting of a histidine, aspartic acid, and catalytic serine (*Xu et al., 2018*). Previous work strongly suggests that the alpha/beta hydrolase domain mainly converts PS to lysoPS (*Kamat et al., 2015*; *Montero-Moran et al., 2010*; *Xu et al., 2018*). The opposing predicted acyltransferase motif in ABHD16A is present in other ABHD family members and some integral ER membrane proteins and has been ascribed various related functions including the ability to either convert single-chain phospholipids, such as lysoPS, back into dual-chain phospholipids or alter the SN2 position of its phospholipid substrates to lysophospholipids/phospholipids (*Aguado and Campbell, 1998*; *Eberhardt et al., 1997*; *Lord et al., 2013*; *Zhao et al., 2008*). We hypothesized that both the acyltransferase and hydrolase domains could contribute enzymatic activities and alter phospholipid shape to promote ER-associated mitochondrial constriction. We therefore generated point mutants within each of these motifs that would be predicted to abrogate enzymatic function: H189A and D194N in the acyltransferase motif (AT mut) and S355A in the alpha/beta hydrolase domain

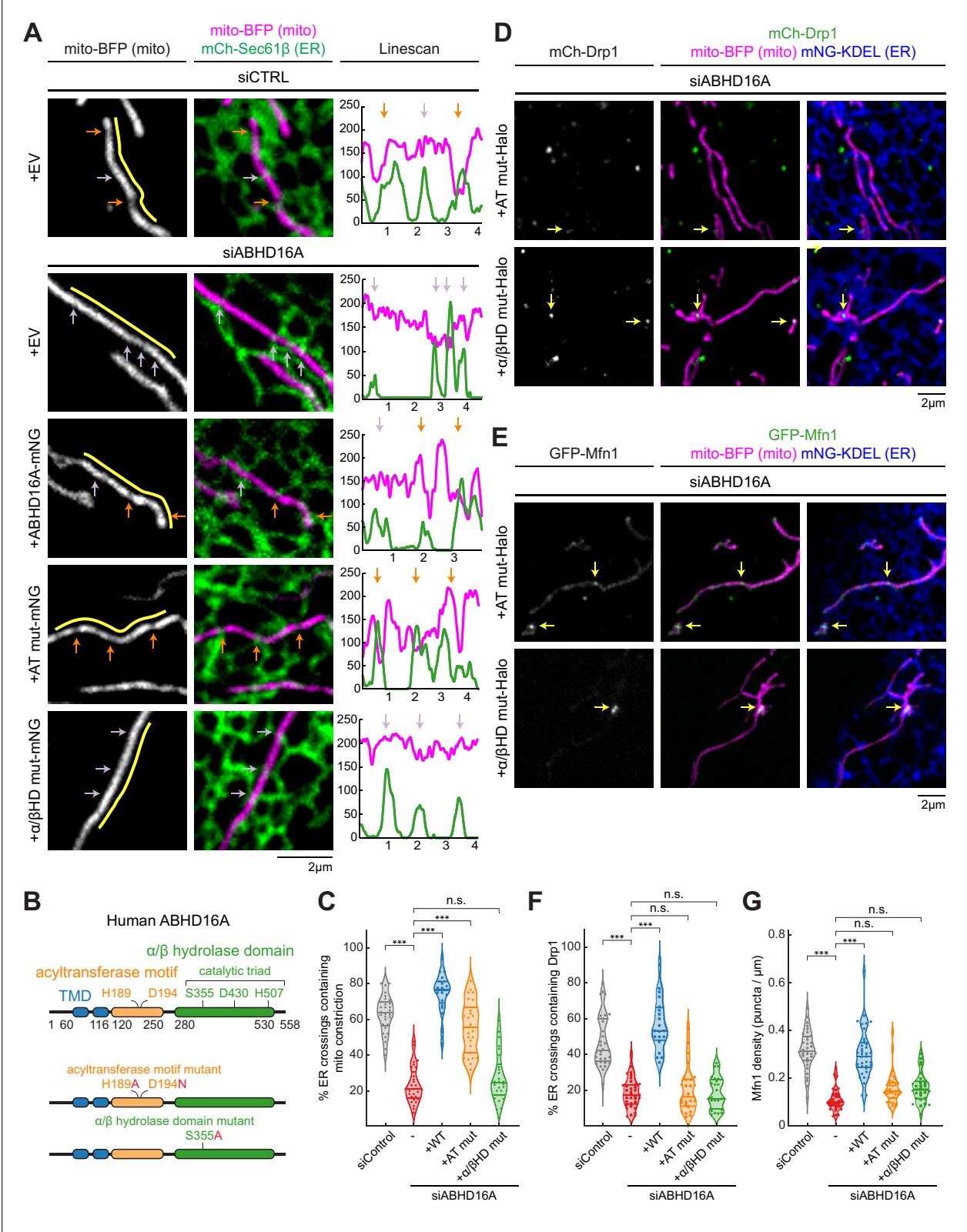

**Figure 6.** Uncoupling ABHD16A's enzymatic requirements during endoplasmic reticulum (ER)-associated mitochondrial constriction, fission, and fusion. (**A**) Representative images and line scans (yellow) of ER-mitochondria crossings in U-2 OS cells transfected with mNG-Sec61β (ER, green), mito-BFP (gray, left; magenta, middle) and either control siRNA (n=30 cells), ABHD16A siRNA (n=29 cells), or ABHD16A siRNA rescued with either wild type (WT) ABHD16A (n=27 cells), the AT mut (acyltransferase motif mutant, n=37 cells), or the α/βHD mut (alpha/beta hydrolase domain mutant, n=26 cells) mNG

*Figure 6 continued on next page*

*Figure 6 continued*

fusion constructs. A representative line scan (yellow, moved aside for visualization purposes) along a mitochondrion (mito-BFP) shows spatial correlation between constrictions (marked by dips in fluorescence intensity, magenta) and ER crossings (green). Orange and purple arrows mark ER crossings with and without constrictions, respectively. (**B**) ABHD16A domain organization with indicated amino acids for each motif or domain (top). Mutations are indicated in red for either the AT or α/βHD mutants. (**C**) Quantification of resolvable ER crossings coincident with a mitochondrial constriction per cell from (**A**). (**D**) As in *Figure 2B*, representative images of U-2 OS cells transfected with mCh-Drp1 (green), mito-BFP (magenta), mNG-Sec61β (ER, blue), and ABHD16A siRNA rescued with either the Halo-tagged ABHD16A AT mut (n=24 cells, top) or α/βHD mut (n=25 cells, bottom). Yellow arrows indicate examples of Drp1 puncta at ER-mitochondria crossings. (**E**) As in *Figure 2A*, representative images of U-2 OS cells transfected with GFP-Mfn1 (green), mito-BFP (magenta), mCh-Sec61β (ER, blue), and ABHD16A siRNA rescued with either the Halo-tagged ABHD16A AT mut (n=33 cells, top) or α/βHD mut (n=28 cells, bottom). Yellow arrows indicate examples of Mfn1 puncta along mitochondria at ER-mitochondria crossings. (**F**) As in *Figure 2E*, quantification of percent ER crossings containing Drp1 puncta from experiments shown in (**D**) and *Figure 2B*. (**G**) As in *Figure 2D*, quantification of Mfn1 density along mitochondrial length from experiments shown in (**E**) and *Figure 2A*. All data were taken from three biological replicates; statistical significance was calculated by one-way ANOVA. n.s., not significant; ***p≤0.001. Scale bar = 2 μm. See *Figure 6—source data 1*.

The online version of this article includes the following source data and figure supplement(s) for figure 6:

**Source data 1.** Related to *Figure 6C, F and G*.

**Figure supplement 1.** ABHD16A's acyltransferase motif and alpha/beta hydrolase domain are required for mitochondrial morphology (related to *Figure 6*).

**Figure supplement 1—source data 1.** Related to *Figure 6—figure supplement 1C*.

**Figure supplement 1—source data 2.** Related to *Figure 6—figure supplement 1G*.

**Figure supplement 1—source data 3.** Related to *Figure 6—figure supplement 1H*.

**Figure supplement 1—source data 4.** Related to *Figure 6—figure supplement 1C*.

**Figure supplement 1—source data 5.** Related to *Figure 6—figure supplement 1C*.

**Figure supplement 1—source data 6.** Related to *Figure 6—figure supplement 1H*.

**Figure supplement 1—source data 7.** Related to *Figure 6—figure supplement 1H*.

---

(α/βHD mut; *Figure 6B*). Cells were depleted with ABHD16A siRNA and were co-transfected with mito-BFP (magenta) and mCh-Sec61β (ER, green) to score ER-associated mitochondrial constriction as before (*Figure 6A* and *Figure 6—figure supplement 1B*). siRNA-resistant WT and mutant ABHD16A constructs were tested for their ability to rescue ER-associated mitochondrial constriction. The acyltransferase motif mutant could rescue ER-associated mitochondrial constriction to a level similar to WT rescue levels, whereas the alpha/beta hydrolase domain mutant could not (*Figure 6A and C*, *Figure 6—figure supplement 1B*, and *Figure 6—figure supplement 1C*). These data suggest that phospholipid hydrolysis from a dual-chain phospholipid to a single-chain phospholipid could be why ABHD16A drives ER-associated mitochondrial constriction.

## ABHD16A's acyltransferase and hydrolase domain are required for fission and fusion

Since the alpha/beta hydrolase activity is sufficient to rescue ER-associated mitochondrial constriction (measured with mito-BFP fluorescence intensity), we expected that it might also rescue fission and fusion node formation. First, we scored ABHD16A motif/domain requirements for fission (Drp1) machinery recruitment to ER-mitochondria MCSs. Each mutant was tested for its ability to rescue Drp1 recruitment in cells depleted of endogenous ABHD16A with siRNA. These cells were co-transfected with siRNA-resistant mutant ABHD16A along with mito-BFP (magenta), mCh-Sec61β (ER, blue), and mCh-Drp1 (green; *Figure 6D*, *Figure 6—figure supplement 1D*, and *Figure 6—figure supplement 1H*). Interestingly, only WT ABHD16A could restore Drp1 recruitment to ER crossings (*Figures 2B, E, 6D and F*, and *Figure 6—figure supplement 1D*). Neither the acyltransferase motif nor alpha/beta hydrolase domain mutants could rescue Drp1 recruitment in ABHD16A-depleted cells (*Figure 6D and F*, and *Figure 6—figure supplement 1D*). In complementary experiments, we also tested whether each mutant could rescue Mfn1 fusion machinery recruitment. Cells were co-transfected with either a control or ABHD16A siRNA, fluorescently tagged Mfn1 (GFP-Mfn1, green), a mito-BFP (magenta), and mNG-Sec61β (ER, blue) and were rescued with siRNA-resistant WT or mutant ABHD16A constructs (ABHD16A-Halo; *Figure 6E* and *Figure 6—figure supplement 1E*). Similar to what was seen for Drp1, only the WT protein could restore Mfn1 accumulation at ER-mitochondria MCSs in ABHD16A-depleted cells (*Figures 2A, D, 6E and G*). Thus, both acyltransferase and alpha/beta hydrolase activities are

required for the recruitment of fission and fusion machinery to ER-associated nodes. Consistent with these results, only the WT ABHD16A can rescue overall mitochondrial morphology (*Figure 6—figure supplement 1F* and *Figure 6—figure supplement 1G*). In addition, the lipase-like motifs (GXSXXG) briefly noted in *Figure 1H* are predicted to function similarly to bacterial lipase motifs (GXSXG). However, we have determined that mutating the predicted catalytic serines of both motifs (S176A and S306A) could still rescue mitochondrial morphology similarly to WT ABHD16A (*Figure 6—figure supplement 1F* and *Figure 6—figure supplement 1G*). We therefore concluded that these lipase-like motifs have either no function or no role in regulating mitochondrial dynamics. So, while the alpha/beta hydrolase domain is sufficient for ER-associated mitochondrial constriction, the acyltransferase motif is still required to form active nodes for fission and fusion. These data suggest that the mechanism of ER-associated mitochondrial node formation is a multistep process that requires several modifications of phospholipids.

## ORP8 is required to deliver ABHD16A-induced altered phospholipids to mitochondria

Since ER-localized ABHD16A is necessary to sustain mitochondrial fission and fusion, we identified potential candidates who could facilitate lipid exchange during ABHD16A-driven mitochondrial node formation. Two published ER-localized lipid transfer proteins were worthy candidates: ORP8, which has been proposed to transfer phospholipids, such as PS, to other organelles at ER-mitochondria and ER-PM MCSs (*Chung et al., 2015*; *Galmes et al., 2016*) and the VPS13A/VPS13D paralogs, which have demonstrated roles in facilitating lipid transfer between the ER and other organelles at MCSs (*Guillén-Samander et al., 2021a*; *Kumar et al., 2018*). To score whether these ER-localized lipid transport proteins contribute to steady state mitochondrial morphology, U-2 OS cells were co-transfected with a mitochondrial matrix marker (mito-BFP, magenta) and either control (siCTRL), OSBPL8 siRNA, or VPS13A/VPS13D siRNAs (which led to efficient protein depletion, *Figure 7F and G*), and mitochondrial morphology was quantified as described previously (*Lee et al., 2016*). Neither ORP8 depletion nor VPS13A/13D depletion by siRNA treatment altered mitochondrial morphology compared to siCTRL-treated cells (*Figure 7A, C, D and E*). We next tested whether ORP8 or VPS13A/D were required for the dramatic mitochondrial fragmentation phenotype observed upon ABHD16A OE. Cells were transfected with either siCTRL, OSBPL8 siRNA, or VPS13A/VPS13D siRNAs and with either EV control (mCherry-N1) or ABHD16A-mCh, as indicated (*Figure 7A, B and C*). ABHD16A-mCh OE promotes mitochondrial fragmentation as previously described (*Figures 4F, 7B and C*). However, mitochondrial morphology is dramatically rescued upon ORP8 depletion and re-fragmented by exogenous siRNA-resistant ORP8 re-expression in the ABHD16AOE/ORP8-depleted cells (*Figure 7B and D*). In contrast, depletion of both VPS13A and VPS13D did not prevent ABHD16A OE-induced fragmentation (*Figure 7C and E*). Thus, ORP8, but not VPS13A/D, facilitates ABHD16A activity during ER-associated mitochondrial constriction. A likely hypothesis is that ABHD16A and ORP8 function to alter the lipid composition at ER-associated mitochondrial constrictions for fission and fusion machinery recruitment.

## Discussion

Through proximity proteomics, we have identified an ER membrane protein, ABHD16A, that regulates the formation of fission and fusion nodes at ER-mitochondria MCSs. ABHD16A localizes to the ER membrane and to a lesser extent mitochondria. Interestingly, only the ER-localized form is necessary and sufficient to maintain mitochondrial morphology. We have shown that ABHD16A is required for mitochondrial constrictions prior to the formation of fission and fusion nodes. Without ABHD16A, nodes are not restored, leading to an overall elongated morphology due to overruling tip-to-tip fusion. Two critical motifs/domains required for these activities are the alpha/beta hydrolase domain, which mainly converts the dual-chain phospholipid, PS, to lysoPS, and the acyltransferase motif, which could convert the inverse reaction. We have identified that only the alpha/beta hydrolase domain is required for IMM constriction, which is required for efficient mitochondrial fission and fusion processes. However, the acyltransferase motif is also required for a step downstream of IMM constriction to allow fission and fusion machineries to accumulate on the OMM at ER-associated nodes. Indeed, both the alpha/beta hydrolase domain and acyltransferase motif are also required to restore mitochondrial

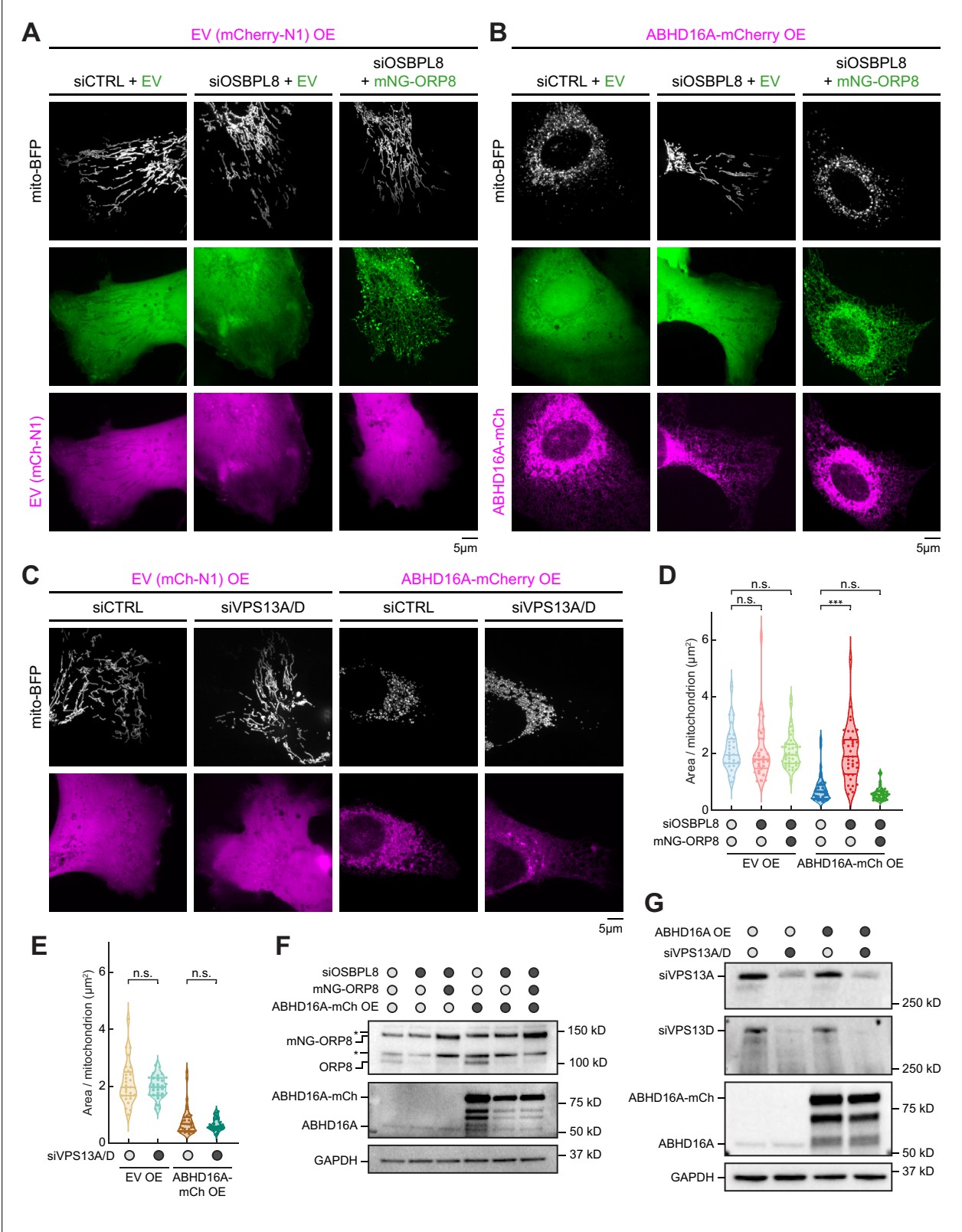

**Figure 7.** ORP8 is required to deliver ABHD16A-induced altered phospholipids to mitochondria. (**A**) Representative images of mitochondrial morphology (labeled by mito-BFP, gray) of U-2 OS cells transfected with empty vector (EV) (mCherry-N1, magenta) and with either siCTRL and mNG-C1 EV (green, n=28 cells, left); OSBPL8 siRNA and mNG-C1 EV (green, n=21 cells, middle); or OSBPL8 siRNA and siRNA-resistant mNG-ORP8 (green, n=41 cells, right). (**B**) Representative images of mitochondrial morphology (labeled by mito-BFP, gray) of U-2 OS cells transfected with

*Figure 7 continued on next page*

*Figure 7 continued*

ABHD16A-mCherry overexpression (OE; magenta) and either siCTRL and mNG-C1 EV (green, n=35 cells, left); OSBPL8 siRNA with mNG-C1 EV (green, n=37 cells, middle); or OSBPL8 siRNA with siRNA-resistant mNG-ORP8 (green, n=37 cells, right). (C) Representative images of mitochondrial morphology (labeled by mito-BFP, gray) of U-2 OS cells transfected with either EV (mCherry-N1, magenta) and siCTRL (n=28 cells); EV and VPS13A/D siRNA (n=30 cells); with ABHD16A-mCherry OE (magenta) and siCTRL (n=36 cells); or with ABHD16A-mCherry OE (magenta) and VPS13A/D siRNA (n=29 cells). (D) Quantification of mean mitochondrial size (area per mitochondrion in $\mu m^2$) within a 15×15 $\mu$m region of interest (ROI) from (A) and (B). (E) Quantification of mean mitochondrial size (area per mitochondrion in $\mu m^2$) within a 15×15 $\mu$m ROI from (C). (F) Representative immunoblot shows efficiency of depletion in U-2 OS cells from (A) and (B) treated with control siRNA or OSBPL8 siRNA and rescued with siRNA-resistant mNG-ORP8. GAPDH serves as a loading control. Asterisk indicates non-specific band. (G) Representative immunoblot shows efficiency of depletion in U-2 OS cells from (C) treated with control siRNA or VPS13A/D siRNA. GAPDH serves as a loading control. All data were taken from three biological replicates; statistical significance was calculated by one-way ANOVA. n.s., not significant; ***$p \leq 0.001$. Scale bar = 5 $\mu$m. See *Figure 7—source data 1*, *Figure 7—source data 2*, *Figure 7—source data 3*, *Figure 7—source data 4*, *Figure 7—source data 5*, *Figure 7—source data 6*, *Figure 7—source data 7*, *Figure 7—source data 8*, *Figure 7—source data 9*.

The online version of this article includes the following source data for figure 7:

**Source data 1.** Related to *Figure 7D and F*.

**Source data 2.** Related to *Figure 7F*.

**Source data 3.** Related to *Figure 7G*.

**Source data 4.** Related to *Figure 7F*.

**Source data 5.** Related to *Figure 7F*.

**Source data 6.** Related to *Figure 7F*.

**Source data 7.** Related to *Figure 7F*.

**Source data 8.** Related to *Figure 7G*.

**Source data 9.** Related to *Figure 7G*.

morphology. Interestingly, yeast encodes a similarly localized and uncharacterized enzyme and potential ABHD16A homolog, YNL320W, containing an alpha/beta hydrolase fold. Future experiments are warranted to test whether at least part of this mechanism can be extended to yeast.

We propose a model whereby ABHD16A's alpha/beta hydrolase domain functions mainly as a PS lipase to build up lysoPS along the mitochondrial membrane for positive membrane curvature. It is possible that the acyltransferase motif subsequently promotes dual-chain phospholipid build up or phospholipid acyl chain modification. One can imagine both single-chain and dual-chain phospholipids are required for positive and negative membrane curvature, respectively, which can promote efficient mitochondrial fission and fusion processes (*Figure 8*, model). Alternatively, several acyltransferase proteins can modify the SN2 position of phospholipids to increase acyl chain asymmetry to potentiate membrane bending and curvature (*Harayama et al., 2014*; *Hishikawa et al., 2014*; *Hofmann, 2000*; *Manni et al., 2018*; *Shindou et al., 2013*; *Zhao et al., 2008*). Through this mechanism, ABHD16A could promote membrane curvature for mitochondrial fission and fusion. Intriguingly, some ABHD family proteins are depalmitoylases, and new evidence suggests ABHD16A has the ability to depalmitoylate some transmembrane proteins (*Cao et al., 2019*; *Lin and Conibear, 2015*; *Shi et al., 2022*). It is also plausible that ABHD16A could depalmitoylate some ER-mitochondria MCS proteins to alter their membrane affinity at nodes.

We have also shown that ER-localized ABHD16A is necessary and sufficient to restore ER-associated mitochondrial morphology and that ABHD16A functions on the ER to affect mitochondrial fission and fusion cycles. Although we cannot visualize any spatial or temporal localization of ABHD16A at ER-mitochondria MCSs, we speculate that an ER tether could facilitate its immobilization at these contacts for ABHD16A's focused activity. ABHD16A could alter lipids on the ER membrane, where a characterized or unidentified lipid transport protein could transport these altered lipids to mitochondria at ER-mitochondria MCSs. Alternatively, it is possible that ABHD16A's cytoplasmic region (~4 nm wide) modifies mitochondrial lipids directly across the MCS bridge. Interestingly, whole cell analysis of S-palmitoylation indicates that ABHD16A contains a palmitoylated cysteine at residue 205 and two predicted palmitoylation sites at residues 284 and 285 (*Thinon et al., 2018*); it will be worthwhile to test whether such modifications could alter its function during ER-associated mitochondrial node formation.

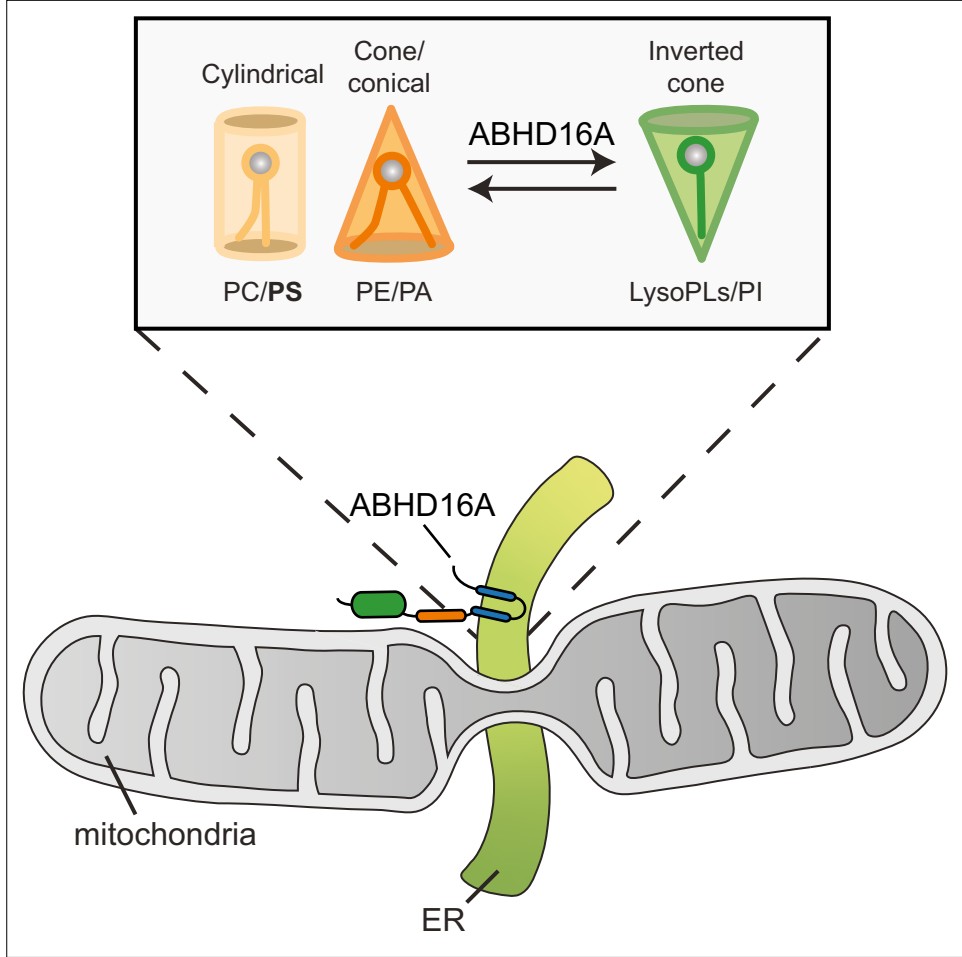

**Figure 8.** Model for how ABHD16A promotes both lysophospholipid and phospholipid formation required for positive and negative membrane curvature to support efficient cycles of fission and fusion.

We have always considered that lipid composition and shape are likely contributors to protein localization and membrane architecture at MCSs, but by what mechanism was unclear. Lipid composition can modulate physicochemical properties of organelle membranes and shape due to the diversity of each phospholipid head group and hydrophobic tail (*Harayama and Riezman, 2018*). For example, phosphatidylethanolamine exists as a cone-shaped lipid, which can create negative spontaneous curvature to promote hemi-fusion intermediates. Cone-shaped lipids have been shown to be required for SNARE-mediated fusion and osteoclast fusion (*Irie et al., 2017*; *Zick et al., 2014*). Dynamin activity is also enhanced by acyl chain asymmetry and polyunsaturation (*Manni et al., 2018*). These data fall in line with ABHD16A converting phospholipids to lysophospholipids and vice versa or altering the acyl chain symmetry to create spontaneous negative and positive membrane curvature necessary for hemi-fusion intermediates at MCSs for constriction, fission, and fusion.

Additionally, many known mitochondrial fission and fusion factors have been associated with specific phospholipids. In vitro and cryoEM studies of Drp1 showed that reconstituted Drp1 prefers to oligomerize around negatively charged phospholipids such as PS and cardiolipin (CL; *Francy et al., 2017*; *Kalia et al., 2018*). In vitro reconstitution of OPA1 and CL membranes is sufficient for tethering and fusion (*Ban et al., 2017*). Interestingly, ABHD16A has the highest specificity for PS (*Kamat et al., 2015*), which constitutes only about 1–2% of the total mitochondrial membrane phospholipid composition (*van Meer et al., 2008*). Thus, it is possible that PS and lysoPS conversion could be secluded to ER-mitochondria membrane contact sites deemed competent for constriction and fission/fusion machinery recruitment. These data also align in an appealing way with Mitofusin 2's proposed capability to cluster PS on mitochondrial membranes (*Hernández-Alvarez et al., 2019*). These data

could suggest that ABHD16A along with an unidentified lipid transport protein or tether induces PS shuttling to allow for Mfn1/2 to localize to nodes. Intriguingly, it is unclear how the adaptor proteins, Mff, Mid49, and Mid51, enrich in punctate localizations at ER-mitochondria contacts prior to Drp1 localization (*Friedman et al., 2011*; *Palmer et al., 2011*). Perhaps lipid alteration and the curvature sensing portions of these adaptor proteins could be responsible for its punctate localization, and it is ABHD16A that creates these lipid changes.

In addition, the ER and mitochondria also form contact sites to transfer phospholipids (*Achleitner et al., 1999*; *Ardail et al., 1993*; *Kornmann et al., 2009*; *Vance, 1990*). Several studies have shown that lipid transfer and mitochondrial morphology are highly interdependent. More specifically, in yeast, the ER-mitochondria encounter structure (ERMES) complex, which is proposed to transfer ER PS and PC to mitochondrial membranes, (*Jeong et al., 2017*; *Kojima et al., 2016*; *Kornmann et al., 2009*) assembles between the ER and mitochondrial membranes at MCSs (*Michel and Kornmann, 2012*; *Murley et al., 2013*). Interestingly, a deletion in the single ERMES complex protein, Gem1, also fragments mitochondria (*Frederick et al., 2004*; *Murley et al., 2013*). In animal cells, Gem1 homologs are Miro1 and Miro2. Miro1/2 proteins regulate mitochondrial and peroxisomal trafficking on microtubules and can elongate both mitochondria and peroxisomes (*Castro et al., 2018*; *Fransson et al., 2006*; *MacAskill et al., 2009a*; *MacAskill et al., 2009b*; *Modi et al., 2019*; *Okumoto et al., 2018*; *Russo et al., 2009*; *Saotome et al., 2008*). The De Camilli lab has shown that the lipid transport protein VPS13D is recruited by Miro (*Guillén-Samander et al., 2021b*), suggesting that lipid transfer must be required for lipid membrane expansion and subsequent elongated mitochondrial morphology. Additionally, recent work from the Nunnari lab also suggests that a buildup of lyso-phosphatidic acid, which promotes positive membrane curvature, sensed by mitochondrial carrier homolog 2 (MTCH2) stimulates mitochondrial fusion (*Labbé et al., 2021*). These data are intriguing because ABHD16A could also promote positive membrane curvature for the formation of fission and fusion nodes. Taken together, these data suggest that lipid transfer and modification at ER-mitochondria MCSs have a direct involvement with mitochondrial elongation and fragmentation by affecting fusion and fission.

## Materials and methods
### DNA plasmids and primer sequences
V5-TurboID-NES was a gift from Alice Ting (Addgene plasmid # 107169; *Branon et al., 2018*). V5-TurboID-Mfn1 and V5-TurboID-Mfn1E209A were generated by subcloning TurboID from V5-TurboID-NES, and AgeI/XhoI sites were used to replace GFP in GFP-Mfn1 or GFP-Mfn1E209 (*Abrisch et al., 2020*). GFP-Mfn1 and GFP-Mfn1E209A were previously described (*Abrisch et al., 2020*). Mito-BFP was previously described (*Friedman et al., 2011*). Mitochondrial-targeting sequence of budding yeast COX4 gene (aa 1–22) was PCR amplified and cloned with XhoI/BamHI sites into mTagBFP-N1. ABHD16A (isoform a, NM_021160.3) was PCR amplified from HeLa cDNA and cloned into HindIII/SacII sites of mNeonGreen-N1 (purchased from Allele Biotechnology) to generate ABHD16A-mNG. mCherry-Sec61β (*Zurek et al., 2011*) subcloned from AcGFP-Sec61β (*Shibata et al., 2008*) into mCherry-C1 using BglII/EcoRI sites. Stim1NTD-ABHD16Acyto-mNG (ABHD16A$^{ER}$) and Tom70NTD-ABHD16Acyto-mNG (ABHD16A$^{mito}$) were generated with Gibson assembly following manufacturer's protocol (New England Biolabs). The templates were ABHD16A-mNG and HeLa cDNA for Stim1's NTD and Tom70's NTD. Primers used to generate these chimeric constructs are: TN378-TN381 for Stim1NTD-ABHD16Acyto-mNG and Tom70NTD-ABHD16Acyto-mNG. SiRNA-resistant ABHD16A constructs were generated from ABHD16A-mNG by QuikChange II site-directed mutagenesis (Aligent Technologies Cat. #200524) following manufacturer's protocol. Primers used were: TN361 and TN362. SiRNA-resistant ABHD16A-mCherry was subcloned using HindIII/SacII sites into mCherry-N1 from siRNA-resistant ABHD16A-mNG. mCherry-Drp1 was previously described (*Friedman et al., 2011*). Drp1 (NM_005690) was PCR amplified from HeLa cDNA and cloned into XhoI/BamHI sites substituting α-Tubulin in mCherry-α-Tubulin. mNG-Sec61β was generated by PCR amplifying human Sec61β from AcGFP-Sec61β (*Shibata et al., 2008*) and inserted into mNG-C1 (purchased from Allele Biotechnology) using HindIII/KpnI sites. mNG-Mfn1 was subcloned using EcoRI/BamHI sites into mNG-N1 from GFP-Mfn1. SiRNA-resistant ABHD16A-Halo was subcloned using HindIII/SacII sites into Halo-N1 from siRNA-resistant ABHD16A-mNG. mMAPLE-OMP25 was a

gift from R. Abrisch. mMAPLE was PCR amplified from mito-mMAPLE, and mMAPLE-C1 vector was derived from AcGFP-C1 by replacing GFP using NheI/BspEI sites. OMP25 was PCR amplified from paGFP-Omp25 (gift from D. Sabatini, Addgene plasmid #69598) and cloned with XhoI/BamHI into mMaple-C1. RA-Sec61β was previously described (*Abrisch et al., 2020*). RA was PCR amplified from RA-NES (gift from R. Campbell, Addgene plasmid #61019; *Ding et al., 2015*), and RA-C1 vector was derived from AcGFP-C1 by replacing GFP using NheI/BspEI sites. Sec61β was PCR amplified and cloned into XhoI/KpnI sites of RA-C1. B-Mff was previously described (*Abrisch et al., 2020*). GB was PCR amplified from GB-NES (gift from R. Campbell, Addgene plasmid #61017; *Ding et al., 2015*) and cloned into NheI/BspEI sites by replacing GFP in GFP-Mff (Addgene plasmid #49153, *Friedman et al., 2011*). SiRNA-resistant ABHD16A-SNAP was subcloned using HindIII/SacII sites into SNAP-N1 from siRNA-resistant ABHD16A-mNG. SiRNA-resistant ABHD16A-mNG lipase-like motifs mutant (S179AS306A), acyltransferase motif mutant (H189AD194N), and alpha/beta hydrolase domain mutant (S355A) were generated from siRNA-resistant ABHD16A-mNG by QuikChange II site-directed mutagenesis (Aligent Technologies Cat. #200524) following manufacturer's protocol. Primers used were: TN347-TN350, TN329-TN332 for each mutation with TN347-TN350 primer pairs used sequentially to generate ABHD16A-mNG lipase-like motifs mutant (S179AS306A). SiRNA-resistant ABHD16A-Halo acyltransferase motif mutant (H189AD194N) and alpha/beta hydrolase domain mutant (S355A) were subcloned using HindIII/SacII sites into Halo-N1 from siRNA-resistant ABHD16A-mNG mutants. ORP8 (isoform a, NM_020841.5) was PCR amplified from HeLa cDNA and cloned into SalI/SacII sites of mNeonGreen-C1 (purchased from Allele Biotechnology) to generate mNG-ORP8. SiRNA-resistant mNG-ORP8 was generated from mNG-ORP8 by QuikChange II site-directed mutagenesis (Aligent Technologies Cat. #200524) following manufacturer's protocol. Primers used were: TN541-542.

## Primers

| Primer name | Primer sequence (5' → 3') |
|---|---|
| TN378 | AGATCTCGAGCTCAagctTATGgatgtatgcgtccgtcttgc |
| TN379 | gggttggtccagcggccGTTCTGGATATAGGCAAACCAG |
| TN380 | CTGGTTTGCCTATATCCAGAACggccgctggaccaaccc |
| TN381 | gcaagacggacgcatacatcCATAagctTGAGCTCGAGATCT |
| TN506 | GACTCAGATCTCGAGCTCAagctTATGGCCGCCTCTAAACCTG |
| TN507 | ggttggtccagcggccCCGGCGCCGTTGC |
| TN508 | GCAACGGCGCCGGGgccgctggaccaacc |
| TN509 | CAGGTTTAGAGGCGGCCATAagctTGAGCTCGAGATCTGAGTC |
| TN361 | ctacttgtacaggaaaggttacttgagtttAAGTaaGgtAgtAccAttttctcactatgctgggacattgctgcta |
| TN362 | tagcagcaatgtcccagcatagtgagaaaaTggTacTacCttACTTaaactcaagtaacctttcctgtacaa gtag |
| TN347 | gagtctcgaggggggccctGcccgccggggtgtggccc |
| TN348 | gggccacacccggcgggCagggccccctcgagactc |
| TN349 | gcccctggaagctggatatGcagtcctgggctggaatca |
| TN350 | tgattccagcccaggactgCatatccagcttccaggggc |
| TN329 | ttcgcccagagcccctgGCccggggggacagcaAacaccctcctcaaccggg |
| TN330 | cccggttgaggagggtgtTtgctgtccccccggGCcaggggctctgggcgaa |
| TN331 | gacatcatcatctacgcctggGccatcggcggcttcactg |
| TN332 | cagtgaagccgccgatggCccaggcgtagatgatgatgtc |
| TN541 | gaagaaaatccttatttccgtttgaaAaaGgtCgtAaaGtggtatttgtcaggattctataaaaagcc |
| TN542 | GGCTTTTTATAGAATCCTGACAAATACCACTTTACGACCTTTTTCAAACGGAAA TAAGGATTTTCTTC |

## General reagents

Mouse anti-V5 tag monoclonal antibody (Thermo Fisher Scientific, Cat# R960-25, RRID: AB_2556564) was used at 1:2000 for western blot and 1:200 for IF.

Rabbit anti-calnexin polyclonal antibody (Enzo Life Sciences, Cat# ADI-SPA-860-F, RRID: AB_11178981) was used at 1:4000 for western blot.

Rabbit anti-calreticulin polyclonal antibody (Abcam, Cat# ab2907, RRID: AB_303402) was used at 1:2000 for western blot.

Rabbit anti-GAPDH antibody (Sigma-Aldrich, Cat# G9545-200UL, RRID: AB_796208) was used at 1:100,000 for western blot.

Mouse anti-tom20 antibody (F-10; Santa Cruz Biotechnology, Cat# sc-17764, RRID: AB_628381) was used at 1:500 for western blot.

Rabbit anti-COX IV monoclonal antibody (Cell Signaling Technology, Cat# 4850 S) was used at 1:1000 for western blot.

Rabbit anti-ABHD16A antibody (Abcam, Cat# ab185549) was used at 1:1000 for western blot.

Rabbit anti-ORP8 polyclonal antibody (GeneTex, Cat# GTX121273) was used at 1:1000 for western blot.

Rabbit anti-VPS13A (Chorein) polyclonal antibody (Novus Biologicals, Cat# NBP1-85641) was used at 1:1000 for western blot.

Rabbit anti-VPS13D polyclonal antibody (Abcam, Cat# ab202285) was used at 1:1000 for western blot.

Donkey anti-mouse IgG (H+L) Highly Cross-Adsorbed Secondary Antibody, Alexa Fluor 647 (Thermo Fisher Scientific, Cat# A-31571, RRID:AB_162542) was used at 1:200 for IF.

Goat anti-rabbit IgG antibody, HRP-conjugate (Sigma-Aldrich, Cat. #12–348) was used at 1:6000 for western blot.

Goat anti-mouse IgG antibody, HRP-conjugate (Sigma-Aldrich, Cat. # 12–349) was used at 1:3000 for western blot.

4–20% Criterion TGX Precast Midi Protein Gels (Bio-Rad, Cat. #5671094 and #5671024) were used to run western blots.

SuperSignal West Pico PLUS Chemiluminescent Substrate (Thermo Fisher, Cat. # 34577) was used to develop western blots.

Standard SDS-PAGE/Western protocols were used to develop western blots.

## Cell culture and transfections

U-2 OS cells (ATCC HTB-96) were cultured in McCoy's 5 A (Invitrogen, Cat# 16600–108) with 10% Fetal Bovine Serum (FBS) (Sigma-Aldrich, Cat# 12,306 C-500ML) and 1% penicillin-streptomycin (Invitrogen, Cat# 15070063) at 37°C with 5% $CO_2$. HeLa cells (ATCC CCL-2) were cultured in DMEM (Gibco, Cat# 12430–062) with 10% FBS (Sigma-Aldrich, Cat# 12,306 C-500ML) and 1% penicillin-streptomycin (Invitrogen, Cat# 15070063) at 37°C with 5% $CO_2$. Cell lines were validated by ATCC upon arrival by short tandem repeat DNA typing using the Promega GenePrint 10 System according to the manufacturer's instructions (Promega #B9510). The cell lines arrived negative for mycoplasma contamination from ATCC.

For imaging, cells were plated on 35 mm imaging dishes (Cellvis, Cat. #D35-10-1.5-N) for 16–20 hr, then transfected for 5 hr with indicated plasmids in 2 mL Opti-MEM (Invitrogen, Cat. #31985–088) using Lipofectamine 3000 transfection kit (Thermo Fisher, Cat. #L3000-150). Cells were imaged 16–20 hr after transfection in FluoroBrite DMEM (Gibco, Cat. #A18967-01) supplemented with 10% FBS, 1% penicillin-streptomycin, 1% GlutaMAX (Gibco, Cat. #35050–061), and 25 mM HEPES. If imaging SNAP-tag or Halo tag, cells were incubated with Janelia Fluor 646, SE (1.5 µM in DMSO) in serum-free FluoroBrite DMEM with added supplements for 30 min prior to imaging or Janelia Fluor X 646 in complete FluoroBrite DMEM with added supplements for 30 min prior to imaging. Janelia Fluor 646, SE, and Janelia Fluor X 646 were gifts from Luke Lavis, Janelia Research Facility.

For transfection, plasmids were incubated at room temperature with 2 µL P3000 per µg of plasmid in 250 µL Opti-MEM, while 2.5 µL/mL (of Opti-MEM) lipofectamine 3000 was also incubated in 250 µL Opti-MEM. After 5 min, plasmids + P3000 and lipofectamine 3000 were mixed together and incubated at room temperature for 20 min. Mixture was added drop-wise onto cells in 1.5 mL Opti-MEM for 5 hr.

For plasmid concentrations, we used 125 ng/mL (for IF) 250 ng/mL (for mass spectrometry) V5-TurboID-Mfn1; 150 ng/mL (for IF) 50 ng/mL (for mass spectrometry) V5-TurboID-Mfn1E209A; 125 ng/mL GFP-Mfn1; 75 ng/mL mito-BFP; 25 ng/mL (low expression) 250 ng/mL (OE) ABHD16A-mNG; 75 ng/mL (KD +rescue) 250 ng/mL (OE) mNG-N1; 200 ng/mL mCherry-Sec61β; 75 ng/mL (KD +rescue) 250 ng/mL (OE) Stim1NTD-ABHD16Acyto-mNG (ABHD16A$^{ER}$); 75 ng/mL (KD + rescue) 250 ng/mL (OE) Tom70NTD-ABHD16Acyto-mNG (ABHD16A$^{mito}$); 75 ng/mL siRNA-resistant ABHD16A-mNG; 100 ng/mL mCherry-N1; 100 ng/mL or 250 ng/mL (OE) siRNA-resistant ABHD16A-mCherry; 75 ng/mL mito-mScarlet; 150 ng/mL mNG-KDEL; 75 ng/mL mCh-Drp1; 200 ng/mL mNG-Sec61β; 125 ng/mL mNG-Mfn1; 75 ng/mL Halo-N1; 75 ng/mL siRNA-resistant ABHD16A-Halo; 150 ng/mL mMAPLE-OMP25; 125 ng/mL RA-Sec61β; 125 ng/mL B-Mff; 75 ng/mL SNAP-N1; 75 ng/mL siRNA-resistant ABHD16A-SNAP; 175 ng/mL siRNA-resistant ABHD16A-mNG lipase-like motifs mutant (S179AS306A); 75 ng/mL siRNA-resistant ABHD16A-mNG acyltransferase motif mutant (H189AD194N); 75 ng/mL siRNA-resistant ABHD16A-mNG alpha/beta hydrolase domain mutant (S355A); 75 ng/mL siRNA-resistant ABHD16A-Halo acyltransferase motif mutant (H189AD194N); 75 ng/mL siRNA-resistant ABHD16A-Halo alpha/beta hydrolase domain mutant (S355A); 125 ng/mL siRNA-resistant mNG-ORP8.

For KD ±rescue experiments, cells were treated with siRNAs twice. The following siRNA oligonucleotides were used: Silencer Negative Control #1 siRNA (Ambion, Cat. #AM4635) used at 25 nM and ABHD16A siRNA (Horizon Discovery, Cat# J-013106-09-0010) used at 25 nM against target sequence: 5'-UGUCCAAAGUGGUGCCGUU-3'. The ABHD16A siRNA used targets a region inside the open reading frame (ORF), so all ABHD16A constructs were mutated to be resistant. OSBPL8 siRNA (Horizon Discovery, Cat# J-009508-07-0005) used at 25 nM against target sequence: 5'-AGAAAGUAGUGAAAUGGUA-3'. The OSBPL8 siRNA used targets a region inside the ORF, so siRNA-resistant mNG-ORP8 was mutated to be resistant. VPS13A SMARTpool siRNA (Horizon Discovery, Cat# L-012878-00-0005) used at 25 nM against target sequences: 5'-GGAUAGAGCUUAUGAUUCA-3', 5'-GAAUGGCACUGGAUAUUAA-3', 5'-UAACACAUCUGCACAUCAA-3', 5'-GCAGCUACAUUCCUCUUAA-3'. VPS13D SMARTpool siRNA (Horizon Discovery, Cat# L-021567-02-0005) used at 25 nM against target sequences: 5'-UCUAAGAACUGCCGAGAAU-3', 5'-CAAGAAAGGCCGAGGUCGA-3', 5'-GGAAGGCAGUGCACGGAAA-3', 5'-AUGUUAAGACUCAGCGAAA-3'. First, cells were plated in 6-well plates (Greiner Bio-One Cellstar, Cat. #657–160) for 16–20 hr then transfected with indicated siRNA oligonucleotides with DharmaFECT 1 Transfection Reagent (Cat. #T-2001–02) for the first round of siRNA treatment. Indicated concentrations of siRNA were incubated in 250 μL serum-free, antibiotic-free media, while 5 μL DharmaFECT 1 Transfection Reagent was incubated in 250 μL serum-free, antibiotic-free media separately at room temperature. After 5 min, siRNA and DharmaFECT 1 mixtures were mixed together and incubated at room temperature for 20 min. Mixture was added drop-wise onto cells into 1.5 mL serum-free, antibiotic-free media for 6 hr. The following day, cells were trypsinized with 0.25% Trypsin-EDTA (Gibco, Cat. #25200–072) and plated into 6-well dishes for immunoblot analysis or 35 mm imaging dishes for 16–20 hr. The second round of transfection was done in Opti-MEM with the same concentration of siRNA oligonucleotides and plasmids necessary for imaging using only lipofectamine 3000 without P3000 for 5 hr. Cell lysate was collected, and images were taken 16–20 hr after transfection. Cell lysate was collected by directly lysing in 2× Laemmli Sample Buffer (BioRad, Cat. #1610737) and boiling for 10 min at 95°C after trypsinizing and washing once with 1× Dulbecco's PBS (Sigma Aldrich, Cat. #D1408-500ML).

## Generating knockout cell lines

Knockout cell lines were generated with CRISPR-Cas9 systems following published protocol (*Ran et al., 2013*). Briefly, guide RNAs were cloned into Lenti-CRISPRv2 (a gift from Feng Zhang, Addgene plasmid # 52961; *Sanjana et al., 2014*), which were then transfected into U-2 OS cells for 16 hr. Cells were recovered for 24 hr and then selected with 2 μg/mL puromycin for 72 hr. Surviving cells were expanded and diluted into single clones. Single clones were expanded and verified for deletion with PCR and immunoblot. The two targeting sequences used for creating ABHD16A KO cells were: CAGATGCCCTGGCACCTCTA and GAGTCCCAGTTGGTCCCTAG.

## Immunofluorescence

HeLa cells were seeded on 35 mm imaging dishes (Cellvis, Cat. #D35-10-1.5-N). 16–20 hr after seeding, cells were transfected with V5-TurboID-Mfn1 or V5-TurboID-Mfn1E209A, GFP-Mfn1, and

mito-BFP. 16–20 hr after transfection, cells were washed once with 1× PBS and fixed with 4% paraformaldehyde in 1× PBS for 15 min. Cells were permeabilized with 0.1% TX100 in 1× PBS for 5 min. Cells were washed three times with 1× PBS and blocked in 10% normal donkey serum and 0.1% TX100 in 1× PBS (blocking buffer) for 60 min. Primary antibody against the V5 tag in blocking buffer was added over night. The next day, cells were washed three times with 1× PBS and stained with secondary antibody in blocking buffer for 30 min. Cells were washed three times with 1× PBS and imaged on the spinning disk confocal.

## Microscopy

All cells were imaged on a spinning disk confocal microscope except for *Figure 3*, which was on a Zeiss LSM 880 Confocal Laser Scanning Microscope equipped with a Plan-Apochromat 63× oil objective (1.4 N.A) and Airyscan detector and controlled by Zen Black (Zeiss). The spinning disk is a Nikon Ti2E body with PSF; CSU-X1 Yokogawa confocal scanner unit; OBIS 405, 488, 561, and 640 lasers and an ANDOR iXON Ultra 512X512 EMCCD camera. Images were acquired with Nikon Plan Apo $\lambda$ 100× oil objective (1.4 N.A). Images were acquired with micro-manager 2.0. Images were processed in Fiji (NIH) and Adobe Illustrator (Adobe).

## Image collection and quantification

For mitochondrial area quantification, z-stacks consisting of 12 serial images were each spaced by 0.2 µm. On Fiji, 15×15 µm ROIs were drawn around resolvable regions of mitochondria to assess morphology. Then, mitochondria were individually counted using the 'Multi-point' tool. The total area of the mitochondria in the ROI was calculated by Otsu thresholding. Thresholded images were then assessed using the 'Analyze Particles' function to obtain the total area in the ROI. Finally, the total mitochondrial area was divided by the number of mitochondria in the ROI to obtain the average area per mitochondrion ($µm^2$).

For ER crossings containing Drp1 puncta, 2 min movies were taken at 5 s intervals. Resolvable ER-mitochondria crossings were marked with the 'Multi-point' tool on Fiji. Presence of a Drp1 puncta were counted (as assessed by puncta present throughout a 2 min movie), and the total number of Drp1 puncta was divided by the total number resolvable ER-mitochondria crossings to get the percentage of ER crossings containing Drp1 puncta.

For Mfn1 density quantifications, three lines were drawn along the length of resolvable mitochondrion per cell using the 'Segmented line' tool in Fiji to obtain mitochondrial length. Then, Mfn1 puncta were counted (as assessed by puncta present throughout a 2 min movie), and the total number of Mfn1 puncta along one mitochondrion was divided by length of the one mitochondrion to give number of Mfn1 puncta per micron (Mfn1 density).

For assessing fission and fusion rate, cells were transfected with mMAPLE-OMP25 as described. Single mitochondria were photoconverted from green to red fluorescence by stimulating with 20 iterations of 100% 405 nm light on the Zeiss LSM 880 Confocal Laser Scanning Microscope. Apparent fusion/fission was scored during live 5 min time-lapse movies with 5 s intervals by the observation of fluorescence mixing or fluorescence separation. The fusion/fission rate was calculated by dividing the number of fusion/fission events per mitochondrion per minute. Mitochondrial fusion was binned in two categories: tip-to-tip and tip-to-middle fusion within these experiments.

For mitochondria overlap with ER or MCSs, 20×20 µm or 15×15 µm ROIs were selected for resolvable regions of ER, mitochondria, and ddFP signal. Then, the 'JaCoP' plugin for Fiji was used to manually threshold each image and calculate Mander's correlation coefficient for the percentage of ER covering mitochondria signal or mitochondria covering ddFP signal.

For mitochondrial constriction line scans and quantifications, z-stacks consisting of 12 serial images that were each spaced by 0.2 µm were taken. Resolvable ER-mitochondria crossings were marked with the 'Multi-point' tool on Fiji. Line scans were performed using Fiji by drawing a line along the mitochondrion, and the fluorescence along the mitochondrion was measured. Constrictions were marked as a ≥40% decrease in mitochondrial signal intensity compared to the neighboring fluorescence peak. The number of mitochondrial constrictions was divided by the total number of resolvable ER-mitochondria crossings to get the percentage ER crossings containing constrictions.

## Statistics

All data presented in all figures are from at least three biological replications. All figure quantifications are represented as violin plots with the bolded line representing the median and the peripheral lines representing quartiles. All statistical analyses were performed in GraphPad Prism 8. When comparing two samples, two-tailed Student's t tests were used. When analyzing significance for more than two samples, one-way ANOVA tests were performed, and p values were derived from Tukey's test. ns = not significant, **p<0.01, ***p<0.001, and ****p<0.0001. Details of statistical analysis and exact values of n (numbers of cells) quantified in each experiment can be found in the figure legends.

## Biotinylation, ER isolation, and biotinylated protein collection

HeLa cells were plated in 20×10 cm dishes ~16–20 hr prior to transfection to attain ~90% confluency the day of biotinylation and sample collection. Cells were transfected with V5-TurboID-Mfn1 or V5-TurboID-Mfn1E209A one day after plating. ~16 hr after transfection, cells were treated with 500 µM biotin for 3 hr. After biotin treatment, cells were washed once with cold 1× PBS, trypsinized, and pelleted. Cell pellets were washed twice with cold 1× PBS and resuspended in 2 m L IB-1 with cOmplete, Mini, EDTA-free Protease Inhibitor Cocktail (Sigma-Aldrich, Cat# 11836170001; 225 mM mannitol, 75 mM sucrose, 30 mM Tris-HCl pH = 7.4, and 0.1 mM EGTA; *Wieckowski et al., 2009*). Cells were lysed via sonication for 10 s three times. Lysed cells were spun at 600× g for 5 min twice to rid of whole cells, debris, and the nuclear fraction. The supernatant was then spun at 7000× g for 20 min twice to rid of the mitochondrial fraction. The supernatant was then spun at 20,000× g for 30 min. The pellet (containing the ER fraction) after the 20,000× g spin was resuspended in IB-2 (225 mM mannitol, 75 mM sucrose, and 30 mM Tris-HCl pH = 7.4) and spun again at 20,000× g for 30 min to rid of other contaminating fractions. The supernatant was taken and spun at 100,000× g for 30 min to pellet the residual ER fraction. The pellet (containing the ER fraction) after the 100,000× g spin was resuspended in IB-2 (225 mM mannitol, 75 mM sucrose, and 30 mM Tris-HCl pH = 7.4) and spun again at 100,000× g for 30 min to rid of other contaminating fractions. The cleaned 20,000 and 100,000× g pellets were resuspended in mitochondria resuspending buffer (MRB, 225 mM mannitol, 75 mM sucrose, 30 mM Tris-HCl pH = 7.4, and 0.1 mM EGTA) and 0.1% SDS, flash frozen in liquid nitrogen, and sent for mass spectrometry.

## Sample preparation for mass spectrometry

All sample preparation and mass spectrometry were conducted at the proteomics core at Sanford Burnham Prebys. Protein concentration was determined using a bicinchoninic acid protein assay (Thermo Scientific). Disulfide bridges were reduced with 5 mM tris (2-carboxyethyl)phosphine at 30°C for 60 min, and cysteines were subsequently alkylated with 15 mM iodoacetamide in the dark at room temperature for 30 min. Affinity purification was carried out in a Bravo AssayMap platform (Agilent) using AssayMap streptavidin cartridges (Agilent). Briefly, cartridges were first primed with 50 mM ammonium bicarbonate and then proteins were slowly loaded onto the streptavidin cartridge. Background contamination was removed with 8 M urea, 50 mM ammonium bicarbonate. Finally, cartridges were washed with Rapid digestion buffer (Promega, Rapid digestion buffer kit), and proteins were subjected to on-cartridge digestion with mass spec grade Trypsin/Lys-C Rapid digestion enzyme (Promega, Madison, WI, USA) at 70°C for 1 hr. Digested peptides were then desalted in the Bravo platform using AssayMap C18 cartridges and dried down in a SpeedVac concentrator.

## LC-MS/MS

Prior to LC-MS/MS analysis, dried peptides were reconstituted with 2% acetonitrile (ACN) and 0.1% formic acid (FA), and concentration was determined using a NanoDropTM spectrophometer (ThermoFisher). Samples were then analyzed by LC-MS/MS using a Proxeon EASY-nanoLC system (ThermoFisher) coupled to a Q-Exactive Plus mass spectrometer (Thermo Fisher Scientific). Peptides were separated using an analytical C18 Aurora column (75 µm × 250 mm, 1.6 µm particles; IonOpticks) at a flow rate of 300 nL/min (60 C) using a 120 min gradient: 1% to 5% B in 1 min, 6% to 23% B in 72 min, 23% to 34% B in 45 min, and 34% to 48% B in 2 min (A=FA 0.1%; B=80% ACN: 0.1% FA). The mass spectrometer was operated in positive data-dependent acquisition mode. MS1 spectra were measured in the Orbitrap in a mass-to-charge (m/z) of 350–1700 with a resolution of 70,000 at m/z 400. Automatic gain control target was set to $1\times10^6$ with a maximum injection time of 100ms. Up

to 12 MS2 spectra per duty cycle were triggered, fragmented by higher-energy C-trap dissociation (HCD), and acquired with a resolution of 17,500 and an AGC target of 5×104, an isolation window of 1.6 m/z, and a normalized collision energy of 25. The dynamic exclusion was set to 20 s with a 10 ppm mass tolerance around the precursor.

### Mass spectrometry data analysis

All mass spectra from were analyzed with MaxQuant software version 1.5.5.1. MS/MS spectra were searched against the *Homo sapiens* Uniprot protein sequence database (downloaded in January 2020) and GPM cRAP sequences (commonly known protein contaminants). Precursor mass tolerance was set to 20 ppm and 4.5 ppm for the first search, where initial mass recalibration was completed and for the main search, respectively. Product ions were searched with a mass tolerance 0.5 Da. The maximum precursor ion charge state used for searching was 7. Carbamidomethylation of cysteine was searched as a fixed modification, while oxidation of methionine and acetylation of protein N-terminal were searched as variable modifications. Enzyme was set to trypsin in a specific mode, and a maximum of two missed cleavages was allowed for searching. The target-decoy-based false discovery rate filter for spectrum and protein identification was set to 1%.

## Acknowledgements

We thank Jonathan Friedman, Rob Abrisch, Robert Campbell, David Sabatini, Alice Ting, and Feng Zhang for sharing plasmids. We thank Luke Lavis for sharing Janelia Fluor 646 SE and Janelia Fluor X 646 for SNAP and Halo tag imaging. We thank Alex Rosa Campos for collecting and analyzing mass spectrometry data. We thank Jonathan Friedman, Eric Sawyer, Jonathan Striepen, Sofia Zaganelli, Bruno Antonny, and Tobias Walther for scientific insight and experimental suggestions. T.T.N. was supported by grants from the NIH (T32 GM008759 and T32 GM142607; GM120998 to G.K.V.). G.K.V. is an investigator of the Howard Hughes Medical Institute.

## Additional information

### Funding

| Funder | Grant reference number | Author |
|---|---|---|
| National Institutes of Health | T32 Training Grants GM008759 and GM142607 | Tricia T Nguyen |
| Howard Hughes Medical Institute | | Gia K Voeltz |
| Howard Hughes Medical Institute | | Tricia T Nguyen |

The funders had no role in study design, data collection and interpretation, or the decision to submit the work for publication.

### Author contributions

Tricia T Nguyen, Conceptualization, Data curation, Formal analysis, Funding acquisition, Validation, Investigation, Visualization, Methodology, Writing - original draft, Writing - review and editing; Gia K Voeltz, Conceptualization, Supervision, Funding acquisition, Writing - review and editing

### Author ORCIDs

Tricia T Nguyen http://orcid.org/0000-0003-2314-7147
Gia K Voeltz http://orcid.org/0000-0003-3199-5402

### Decision letter and Author response

Decision letter https://doi.org/10.7554/eLife.84279.sa1
Author response https://doi.org/10.7554/eLife.84279.sa2

# Additional files

## Supplementary files
• MDAR checklist

## Data availability
All data generated or analyzed during this study are included in the manuscript and supporting files. Source Data files have been provided for Figures 1-7 and Figure supplements 1, 2, 4, 5, and 6. Source data contains numerical data or either uncropped western blots used to generate the figures.

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
