## [Editor Report]

The authors have used state-of-the-art tools to discover and visualize the role of a known ER-localized lipid hydrolase/acyl transferase in creating lipids that facilitate the localization of proteins required for mitochondrial fission and fusion at nodal points of interaction between the ER and mitochondria. The data are clear, quantitative, and compelling with respect to the role of this protein in the processes of mitochondrial constriction, fission, and fusion.

---

## [Decision Letter]

**Decision letter after peer review:**

Thank you for submitting your article "An ER phospholipid hydrolase drives ER-associated mitochondrial constriction for fission and fusion" for consideration by *eLife*. Your article has been reviewed by 3 peer reviewers, including Randy Schekman as Reviewing Editor and Reviewer #1, and the evaluation has been overseen by Benoît Kornmann as the Senior Editor.

The reviewers have discussed their reviews with one another, and the Reviewing Editor has drafted this to help you prepare a revised submission. Our recommendation is to consider and respond to the comments but we feel the work can be published with appropriate changes to the text alone.

Essential revisions:

1) 1. The data in Figure 1 G are very convincing but we are left wondering about the identity of the other even more highly Dnm1-dependent ER-enriched proteins mentioned in line 98 on p. 4? How many of these 18 more highly enriched proteins are predicted to be integral ER membrane proteins and among them are any known enzymes or proteins that function in other processes? The absence of any proteomic data on this experiment was a little unsatisfying. Do the authors wish to reserve these others for future analysis or is there some other less interesting explanation for the absence of any discussion? Does *eLife* policy now require the complete disclosure of all such data?

2. Do yeast species have an Aphid homolog and if so, has its function in mitochondrial morphology or anything else such as ER shape change been noted?

3. The discussion of the organization of Aphyd in the ER membrane is incomplete and confusing. From Figure 1H we see that the membrane anchor domain is near the N-terminus but we are not given an explicit statement of the orientation of the protein with respect to the ER bilayer. I assume the C-terminal large domain is oriented to the cytoplasm but has this actually been established in published work? If not, I think establishing the topology of the protein is important for their work.

The cartoon in Figure 8 adds confusion to this issue. Here the ER membrane proximal acyl transferase domain is shown in the cytoplasm but the large, presumably very C-terminal hydrolase domain is shown embedded into the mitochondrial envelope, possibly even the cristae membrane. Surely the authors do not mean this. Figure 8 should be clarified.

4. It seems unnecessary to rename a protein that has already been studied, especially when the new name doesn't inform on a new function but rather is merely a collection of a different set of letters from the old name. This is not renaming of an uncharacterized, unstudied generically named gene (i.e., TMEM, KIAA, FAM, cXorfY, etc.), and it is not naming a gene with a phenotype from a genetic screen. ABHD16A is part of the aptly named ABHD family (α/β hydrolase domain). Further, other labs, in numerous publications, have previously studied the phospholipid hydrolase activity of ABHD16A and analyzed phenotypes and mechanisms using KO cells and mice in detail, as appropriately cited by the authors (e.g., Singh et al., Kamat et al.). Making an unneeded (and still pretty vague) extra name for this protein might ultimately confuse people and make it harder to search the literature without adding any extra obvious value. On the contrary, it turns a name that describes a relevant hydrolase fold (ABHD16A) into a neologism that, likely by design, is a homonym of (and conjures up the image of) an insect. To avoid confusion in the field, the authors may wish to consider keeping the original ABHD16A name.

5. This initial study conducts a rigorous analysis of the role of newly-identified Aphyd in controlling mitochondrial fission/fusion but does not provide any biochemical insights into its lipid-modifying activities – indeed, one can't expect to solve everything in one paper! This question seems complicated since Aphyd contains domains that could catalyze opposite reactions – conversion of a phospholipid such as PS to its lyso- version by a hydrolase activity and conversion of a lyso-phospholipid such as lyso-PS back to a two-chain phospholipid. The analysis of mutants suggests both activities may be needed, but in different ways. In the absence of further biochemical analysis, this last part of the study seems quite speculative to me and could be toned down. Related to this point, in absence of any data showing the lipid specificity of Aphyd activity, the generic fusion/fission model in Figure 8 doesn't add much and the paper would be fine without it.

6. Compared to the hydrolase domain, which has been well characterized biochemically by others to be a PS lipase (based on in vitro assays and lipidomics data from KO cells), the acyltransferase domain is much less well characterized. Overall, published data on ABHD16A KO cells suggest a primary role for this enzyme as a PS lipase. These prior findings are not appropriately reflected in the interpretations and discussions in this manuscript, and this is a particularly important omission to correct given the strong and specific implication of PS in this pathway as shown by the authors' compelling ORP8 data. The authors are encouraged to clarify their model to better take into consideration what is known about ABHD16A's role as a PS lipase.

7. The authors do not address how ABHD16A might have its activity regulated to only produce lipids at these sites. It is a constitutive ER-resident protein, and their model would be strengthened if a proposal and/or supporting evidence could be provided to answer some of the questions put forth by their model, including the following:

Does ABHD16A activity change local levels of PS or lysoPS within these organelles? PS levels could be addressed using imaging biosensors, whereas organelle-specific lipidomics is admittedly more challenging.

Are lipids proposed to be generated in cis on the ER by ABHD16A and transferred to the mitochondria or is lipid hydrolysis proposed to occur in trans? If the latter, the dimensions of the contact sites and the tethered enzyme consistent with the model?

8. Despite ABHD16A being the main focus of this manuscript, there are limited analyses of its spatial and temporal localization.

In Figure 1, it was identified by the difference between the Mfn1 and Mfn1E209A. Does ABHD16A co-localize with the Mfn1 puncta?

Based on Figure 2 and Figure 5, a hierarchy of "MCS formation  ABHD16A activity  Drp1/Mfn1 recruitment" can be inferred. Live-cell imaging would help authors to confirm this hierarchy. If ABHD16A does not accumulate in MCS before fission/fusion, what is mentioned in the comment #2 becomes even more important.

9. Three hours seems like a long time for proximity biotinylation with the optimized TurboID ligase, which is typically used in much shorter pulses (~10 min). The authors are encouraged to comment in the manuscript on the need for such long labeling times.

10. What do the "lipase-like motifs" indicated in gray in Figure 1H correspond to? Are they residues H189 and S355 in Figure 6B? The authors should explain more about what they mean by this phrase, as it is confusing that one of them exists in the acyltransferase domain and the other in the α/β hydrolase domain.

11. The rescue presented throughout is convincing, but all phenotypes derive from a single ABHD16A siRNA. Inclusion of a second siRNA + rescue for a select subset of experiments would help to increase the rigor of the phenotypic findings.

12. Line 250: Figure S6A-B callout should be S5A-B.

13. The data presentation in Figure 3F is confusing, with numbers not adding up (by design but still kind of misleading). I recommend that the authors present it in a different manner, perhaps with n values in the legend and numbers in the graph representing the values in the shaded graph bars? The point is that ABHD16A KD lowers tip-to-mid fusion the most, and rescue can overshoot, but the provided numbers are confusing.

*Reviewer #1 (Recommendations for the authors):*

Nguyen and Voeltz have made a valuable contribution in the finding that Mfn1-Turbo-ID identified a known ER phospholipid hydrolase/acyl transferase as a key player in organizing the nodes of interaction between the ER and mitochondria. I found the experiments well-designed, logical, nicely quantitative and the conclusions convincing.

I have just a few small issues that could be better explained or developed and one general question that would be good to discuss among the set of reviewers.

1. The data in Figure 1 G are very convincing but we are left wondering about the identity of the other even more highly Dnm1-dependent ER-enriched proteins mentioned in line 98 on p. 4? How many of these 18 more highly enriched proteins are predicted to be integral ER membrane proteins and among them are any known enzymes or proteins that function in other processes? The absence of any proteomic data on this experiment was a little unsatisfying. Do the authors wish to reserve these others for future analysis or is there some other less interesting explanation for the absence of any discussion? Does *eLife* policy now require the complete disclosure of all such data?

2. Do yeast species have an Aphid homolog and if so, has its function in mitochondrial morphology or anything else such as ER shape change been noted?

3. The discussion of the organization of Aphyd in the ER membrane is incomplete and confusing. From Figure 1H we see that the membrane anchor domain is near the N-terminus but we are not given an explicit statement of the orientation of the protein with respect to the ER bilayer. I assume the C-terminal large domain is oriented to the cytoplasm but has this actually been established in published work? If not, I think establishing the topology of the protein is important for their work.

The cartoon in Figure 8 adds confusion to this issue. Here the ER membrane proximal acyl transferase domain is shown in the cytoplasm but the large, presumably very C-terminal hydrolase domain is shown embedded into the mitochondrial envelope, possibly even the cristae membrane. Surely the authors do not mean this. Figure 8 should be clarified.

4. Issue for discussion: Aphyd is shown to be broadly distributed along the ER membrane but the question remains are the lysophosphatides and reacylated phospholipids made at points of contact with mitochondria? Or are these phospholipids formed randomly and then phase separate to form the points of contact with mitochondria? Could Aphyd be regulated by contact to activate the hydrolase domain to initiate mitochondrial constriction or by contact with the full C-terminal domain to activate both enzyme activities? This may be an appropriate starting point for a subsequent investigation but one deficiency in this paper is the absence of any enzymatic activity assays to detect the production of lysophosphatides or reacylated phospholipids dependent on membranes engaged in the full range of ER-mitochondrial interactions.

*Reviewer #2 (Recommendations for the authors):*

1. I object to Aphyd as the name. This is not renaming of an uncharacterized, unstudied generically named gene (i.e., TMEM, KIAA, FAM, cXorfY, etc.), and it is not naming a gene with a phenotype from a genetic screen. ABHD16A is part of the aptly named ABHD family (α/β hydrolase domain). Further, other labs, in numerous publications, have previously studied the phospholipid hydrolase activity of ABHD16A and analyzed phenotypes and mechanisms using KO cells and mice in detail, as appropriately cited by the authors (e.g., Singh et al., Kamat et al.). Making an unneeded (and still pretty vague) extra name for this protein might ultimately confuse people and make it harder to search the literature without adding any extra obvious value. On the contrary, it turns a name that describes a relevant hydrolase fold (ABHD16A) into a neologism that, likely by design, is a homonym of (and conjures up the image of) an insect.

2. Three hours seems like a long time for proximity biotinylation with the optimized TurboID ligase, which is typically used in much shorter pulses (~10 min). The authors are encouraged to comment in the manuscript on the need for such long labeling times.

3. What do the "lipase-like motifs" indicated in gray in Figure 1H correspond to? Are they residues H189 and S355 in Figure 6B? The authors should explain more about what they mean by this phrase, as it is confusing that one of them exists in the acyltransferase domain and the other in the α/β hydrolase domain.

4. The rescue presented throughout is convincing, but all phenotypes derive from a single ABHD16A siRNA. Inclusion of a second siRNA + rescue for a select subset of experiments would help to increase the rigor of the phenotypic findings.

5. Line 250: Figure S6A-B callout should be S5A-B.

6. The data presentation in Figure 3F is confusing, with numbers not adding up (by design but still kind of misleading). I recommend that the authors present it in a different manner, perhaps with n values in the legend and numbers in the graph representing the values in the shaded graph bars? The point is that ABHD16A KD lowers tip-to-mid fusion the most, and rescue can overshoot, but the provided numbers are confusing.

*Reviewer #3 (Recommendations for the authors):*

I am positive about this study but do have a couple of suggestions.

1. It seems unnecessary to rename a protein that has already been studied, especially when the new name doesn't inform on a new function but rather is merely a collection of a different set of letters from the old name. To avoid confusion in the field, the authors may wish to consider keeping the original ABHD16A name.

2. This initial study conducts rigorous analysis of the role of newly-identified Aphyd in controlling mitochondrial fission/fusion but does not provide any biochemical insights into its lipid-modifying activities – indeed, one can't expect to solve everything in one paper! This question seems complicated since Aphyd contains domains that could catalyze opposite reactions – conversion of a phospholipid such as PS to its lyso- version by a hydrolase activity and conversion of a lyso-phospholipid such as lyso-PS back to a two-chain phospholipid. The analysis of mutants suggests both activities may be needed, but in different ways. In the absence of further biochemical analysis, this last part of the study seems quite speculative to me and could be toned down. Related to this point, in absence of any data showing the lipid specificity of Aphyd activity, the generic fusion/fission model in Figure 8 doesn't add much and the paper would be fine without it.

---

## [Author Response]

Essential revisions:1) 1. The data in Figure 1 G are very convincing but we are left wondering about the identity of the other even more highly Dnm1-dependent ER-enriched proteins mentioned in line 98 on p. 4? How many of these 18 more highly enriched proteins are predicted to be integral ER membrane proteins and among them are any known enzymes or proteins that function in other processes? The absence of any proteomic data on this experiment was a little unsatisfying. Do the authors wish to reserve these others for future analysis or is there some other less interesting explanation for the absence of any discussion? Does eLife policy now require the complete disclosure of all such data?

The raw proteomics data is presented in Figure 1 source data 4. However, we now include a list of the top 50 proteins sorted by high enrichment in the Mfn1 WT sample and low enrichment (zero) in the Mfn1 E209A mutant sample in Figure S1 (also discussed in the text on line 99-103). The top 18 proteins, prior to ABHD16A, are either not localized to the ER or in the case of ARL6IP1, is characterized as an ER-shaping protein (Yamamoto Y, et al. Biochem J. 2014). The method of ER membrane enrichment described in the methods is, unfortunately, not perfect. We can identify ER membrane enrichment via calnexin, reticulon-4, and OSBPL-8, in both the WT and the mutant samples; but no known proteins were enriched in only the WT sample that function at ER-mitochondria MCSs. However, we did see that fission machinery Drp1, which is known to localize at nodes containing Mfn1, enriched highly in the Mfn1 WT sample compared to the Mfn1 E209A sample in a separate fractionation by mass spectrometry (data not included).

2. Do yeast species have an Aphid homolog and if so, has its function in mitochondrial morphology or anything else such as ER shape change been noted?

It is possible that yeast contain an ABHD16A functional homolog, YNL320W. In some preliminary work this protein is localized to the ER and to a lesser extent, mitochondria, which is notably similar to ABHD16A. It has no sequence homology, however, when analyzed in Pfam for domain homology, it is also predicted to contain a possible α/β hydrolase fold. This protein has not been studied in the literature. This information is now mentioned in the discussion in lines 399-402.

3. The discussion of the organization of Aphyd in the ER membrane is incomplete and confusing. From Figure 1H we see that the membrane anchor domain is near the N-terminus but we are not given an explicit statement of the orientation of the protein with respect to the ER bilayer. I assume the C-terminal large domain is oriented to the cytoplasm but has this actually been established in published work? If not, I think establishing the topology of the protein is important for their work.

We apologize for not mentioning that ABHD16A’s C-terminal domain has been reported to face the cytoplasm. These data are reported in Supplementary Figure 15 of Kamat SS, et al. Nat Chem Biol. 2015. Singh S, et al. Biochemistry. 2020, also reports that ABHD16A enriches in the ER fraction and to a lesser extent, mitochondria, in Figure 2A and 2B in mouse tissue and the Neuro-2a cell line. These data are now mentioned in the text in lines 110 and 133-134.

The cartoon in Figure 8 adds confusion to this issue. Here the ER membrane proximal acyl transferase domain is shown in the cytoplasm but the large, presumably very C-terminal hydrolase domain is shown embedded into the mitochondrial envelope, possibly even the cristae membrane. Surely the authors do not mean this. Figure 8 should be clarified.

We agree that the cartoon is somewhat confusing. We speculate that ABHD16A’s C-terminal region could either alter lipids on the ER membrane, where an unidentified lipid transport protein can transport these altered lipids to mitochondria; or ABHD16A’s C-terminal region could alter mitochondrial lipids directly in trans. We have changed and simplified the cartoon and moved ABHD16A’s C-terminal domain to the cytoplasm.

4. It seems unnecessary to rename a protein that has already been studied, especially when the new name doesn't inform on a new function but rather is merely a collection of a different set of letters from the old name. This is not renaming of an uncharacterized, unstudied generically named gene (i.e., TMEM, KIAA, FAM, cXorfY, etc.), and it is not naming a gene with a phenotype from a genetic screen. ABHD16A is part of the aptly named ABHD family (α/β hydrolase domain). Further, other labs, in numerous publications, have previously studied the phospholipid hydrolase activity of ABHD16A and analyzed phenotypes and mechanisms using KO cells and mice in detail, as appropriately cited by the authors (e.g., Singh et al., Kamat et al.). Making an unneeded (and still pretty vague) extra name for this protein might ultimately confuse people and make it harder to search the literature without adding any extra obvious value. On the contrary, it turns a name that describes a relevant hydrolase fold (ABHD16A) into a neologism that, likely by design, is a homonym of (and conjures up the image of) an insect. To avoid confusion in the field, the authors may wish to consider keeping the original ABHD16A name.

We have considered this and can understand the confusion. We have changed Aphyd back to ABHD16A. Thank you.

5. This initial study conducts a rigorous analysis of the role of newly-identified Aphyd in controlling mitochondrial fission/fusion but does not provide any biochemical insights into its lipid-modifying activities – indeed, one can't expect to solve everything in one paper! This question seems complicated since Aphyd contains domains that could catalyze opposite reactions – conversion of a phospholipid such as PS to its lyso- version by a hydrolase activity and conversion of a lyso-phospholipid such as lyso-PS back to a two-chain phospholipid. The analysis of mutants suggests both activities may be needed, but in different ways. In the absence of further biochemical analysis, this last part of the study seems quite speculative to me and could be toned down. Related to this point, in absence of any data showing the lipid specificity of Aphyd activity, the generic fusion/fission model in Figure 8 doesn't add much and the paper would be fine without it.

We have toned down the speculation about the acyltransferase motif, since there has been no previous biochemical characterization of this region. We also tone down several strong statements that phospholipid hydrolysis is the mechanism. These changes can be found in lines 112-119, 300-311, and 404-412. We include more discussion about the potential model or function, citing the relevant papers. The different models are discussed in lines 402-418.

We have also clarified the model by including ABHD16A’s predicted phospholipid activity. If the reviewers prefer, we can remove the model entirely.

6. Compared to the hydrolase domain, which has been well characterized biochemically by others to be a PS lipase (based on in vitro assays and lipidomics data from KO cells), the acyltransferase domain is much less well characterized. Overall, published data on ABHD16A KO cells suggest a primary role for this enzyme as a PS lipase. These prior findings are not appropriately reflected in the interpretations and discussions in this manuscript, and this is a particularly important omission to correct given the strong and specific implication of PS in this pathway as shown by the authors' compelling ORP8 data. The authors are encouraged to clarify their model to better take into consideration what is known about ABHD16A's role as a PS lipase.

We have edited the introduction, results, and discussion to represent that ABHD16A has been characterized as mainly a PS lipase. There is solely conservation data that many ABHD family proteins contain both an acyltransferase motif as well as a hydrolase domain. It is not clear whether ABHD16A has a functional acyltransferase activity. These changes can be found in lines 105-107, 112-119, 304-311, and 404-414.

7. The authors do not address how ABHD16A might have its activity regulated to only produce lipids at these sites. It is a constitutive ER-resident protein, and their model would be strengthened if a proposal and/or supporting evidence could be provided to answer some of the questions put forth by their model, including the following:Does ABHD16A activity change local levels of PS or lysoPS within these organelles? PS levels could be addressed using imaging biosensors, whereas organelle-specific lipidomics is admittedly more challenging.

We have considered using a biosensor, such as GFP-LactC2; however, PS sensors have high intensity at the plasma membrane, making it difficult to identify the 1-2% of PS on mitochondria by this method. We have also done lipidomics on purified mitochondria of WT vs. ABHD16A KO U-2 OS cells with Lipotype and do not see any differences between the two conditions. However, upon discussion with labs familiar with lipidomics, it is quite difficult to detect “decent” levels of PS and lysoPS, especially on mitochondria, because it is such a small pool of the total lipids.

Are lipids proposed to be generated in cis on the ER by ABHD16A and transferred to the mitochondria or is lipid hydrolysis proposed to occur in trans? If the latter, the dimensions of the contact sites and the tethered enzyme consistent with the model?

We believe that there could be several models for ABHD16A’s mechanism. We propose the following:

1. ABHD16A can alter lipids on the ER membrane, where an unidentified lipid transport protein or known lipid transport protein can transport these altered lipids to mitochondria

2. ABHD16A’s C-terminal region could alter mitochondrial lipids directly in trans

a. ABHD16A C-terminal region is predicted to be about 4nm wide. While it is possible that it could bridge across the MCS gap to modify lipids on mitochondria, it seems too short because MCSs are generally 10-30nm by EM measurements.

These points have been added to the discussion in lines 420-431.

8. Despite ABHD16A being the main focus of this manuscript, there are limited analyses of its spatial and temporal localization.In Figure 1, it was identified by the difference between the Mfn1 and Mfn1E209A. Does ABHD16A co-localize with the Mfn1 puncta?Based on Figure 2 and Figure 5, a hierarchy of "MCS formation  ABHD16A activity  Drp1/Mfn1 recruitment" can be inferred. Live-cell imaging would help authors to confirm this hierarchy. If ABHD16A does not accumulate in MCS before fission/fusion, what is mentioned in the comment #2 becomes even more important.

Even under low expression of ABHD16A, we do not see any spatial or temporal colocalization with Mfn1 puncta. We speculate that an ER tether could facilitate its immobilization at these contact sites, for focused activity. Future studies need to be done of endogenous tagging, single particle tracking, and/or ABHD16A immunoprecipitation to find the tether or lipid transfer protein responsible for its focused activity. Interestingly, the Zorzano lab has suggested that Mfn2 clusters PS on mitochondria (Hernandez-Alvarez MI, et al. Cell. 2019). These data suggest an appealing possibility which is that ABHD16A along with an unidentified lipid transport protein or tether could work with Mfn1/2 to induce PS clustering at nodes. These speculations have been added to the discussion in lines 422-426 and 450-460.

9. Three hours seems like a long time for proximity biotinylation with the optimized TurboID ligase, which is typically used in much shorter pulses (~10 min). The authors are encouraged to comment in the manuscript on the need for such long labeling times.

Upon initial testing of transient transfection of TurboID-Mfn1 constructs, we noticed that we could only see robust biotinylation profiles assayed using streptavidin-HRP at 3 hours. We have added this information in the Results section stated as:

“TurboID proximity biotinylation experiments were performed by transfecting HeLa cells with very low levels to express near endogenous levels of either exogenous V5-TurboID-Mfn1 or V5-TurboID-Mfn1E209A followed by treatment with 500 µM biotin for 3 hours (Figure 1C). Three hours was the minimal time frame to produce robust biotinylation profiles upon expression of low levels of each construct (Figure 1F).”

10. What do the "lipase-like motifs" indicated in gray in Figure 1H correspond to? Are they residues H189 and S355 in Figure 6B? The authors should explain more about what they mean by this phrase, as it is confusing that one of them exists in the acyltransferase domain and the other in the α/β hydrolase domain.

Lipase motifs in bacteria contain motifs of GXSXG. In Xu J, et al. Open Biol. 2018, they report that ABHD16A contains motifs similar to these lipase motifs, termed “lipase-like motifs” that consist of a GXSXXG, now noted in Figure 1H. Upon mutation of the serine, we did not observe any changes in mitochondrial morphology compared to ABHD16A knockdown + re-expression of WT ABHD16A. We believe that these motifs are non-functional at least in relation to the mitochondrial phenotypes we assess.

Because the lipase domain was not required for any rescue functions, we chose to remove it from the main figures so that they were not too busy. However, we have added that the lipase-like motifs mutant (LLM mut, S176AS306A) rescues mitochondrial morphology similarly to WT ABHD16A in an ABHD16A depletion background in Figure S6.

We have added this information and these data into the manuscript in the Results section in lines 119-122 for Figure 1 and mentioned again in lines 346-351 for Figure 6.

11. The rescue presented throughout is convincing, but all phenotypes derive from a single ABHD16A siRNA. Inclusion of a second siRNA + rescue for a select subset of experiments would help to increase the rigor of the phenotypic findings.

We had also generated an analyzed an ABHD16A KO U-2 OS cell line that has the same phenotypes as ABHD16A depleted cells. These data are now shown in Figure S4F and S4G.

12. Line 250: Figure S6A-B callout should be S5A-B.

This has been changed in the manuscript.

13. The data presentation in Figure 3F is confusing, with numbers not adding up (by design but still kind of misleading). I recommend that the authors present it in a different manner, perhaps with n values in the legend and numbers in the graph representing the values in the shaded graph bars? The point is that ABHD16A KD lowers tip-to-mid fusion the most, and rescue can overshoot, but the provided numbers are confusing.

We agree and have changed the presentation of these data in Figure 3F. The data now displayed is a table per condition of the following: total number of mitochondria counted, total fusion events counted, total tip-to-middle fusion events counted, total tip-to-tip fusion events counted, and then a comparison of the different rates of fusion with and without ABHD16A for: all fusion, tip-to-middle fusion and tip-to-tip fusion. We hope this clarifies that tip-to-middle fusion events are mainly affected. These changes are also present in lines 195-202.

Reviewer #1 (Recommendations for the authors):Nguyen and Voeltz have made a valuable contribution in the finding that Mfn1-Turbo-ID identified a known ER phospholipid hydrolase/acyl transferase as a key player in organizing the nodes of interaction between the ER and mitochondria. I found the experiments well-designed, logical, nicely quantitative and the conclusions convincing.I have just a few small issues that could be better explained or developed and one general question that would be good to discuss among the set of reviewers.1. The data in Figure 1 G are very convincing but we are left wondering about the identity of the other even more highly Dnm1-dependent ER-enriched proteins mentioned in line 98 on p. 4? How many of these 18 more highly enriched proteins are predicted to be integral ER membrane proteins and among them are any known enzymes or proteins that function in other processes? The absence of any proteomic data on this experiment was a little unsatisfying. Do the authors wish to reserve these others for future analysis or is there some other less interesting explanation for the absence of any discussion? Does eLife policy now require the complete disclosure of all such data?2. Do yeast species have an Aphid homolog and if so, has its function in mitochondrial morphology or anything else such as ER shape change been noted?3. The discussion of the organization of Aphyd in the ER membrane is incomplete and confusing. From Figure 1H we see that the membrane anchor domain is near the N-terminus but we are not given an explicit statement of the orientation of the protein with respect to the ER bilayer. I assume the C-terminal large domain is oriented to the cytoplasm but has this actually been established in published work? If not, I think establishing the topology of the protein is important for their work.The cartoon in Figure 8 adds confusion to this issue. Here the ER membrane proximal acyl transferase domain is shown in the cytoplasm but the large, presumably very C-terminal hydrolase domain is shown embedded into the mitochondrial envelope, possibly even the cristae membrane. Surely the authors do not mean this. Figure 8 should be clarified.

These points have been addressed above.

4. Issue for discussion: Aphyd is shown to be broadly distributed along the ER membrane but the question remains are the lysophosphatides and reacylated phospholipids made at points of contact with mitochondria? Or are these phospholipids formed randomly and then phase separate to form the points of contact with mitochondria? Could Aphyd be regulated by contact to activate the hydrolase domain to initiate mitochondrial constriction or by contact with the full C-terminal domain to activate both enzyme activities? This may be an appropriate starting point for a subsequent investigation but one deficiency in this paper is the absence of any enzymatic activity assays to detect the production of lysophosphatides or reacylated phospholipids dependent on membranes engaged in the full range of ER-mitochondrial interactions.

These points have been addressed above. It is definitely quite tricky to assess lipids via organelle lipidomics. Future studies would definitely delve into the mechanism of ABHD16A with biochemical assays. As mentioned above, we speculate that an ER tether could facilitate its immobilization at these contact sites, for focused activity. Future studies need to be done of endogenous tagging, single particle tracking, and/or ABHD16A immunoprecipitation to find the tether or lipid transfer protein responsible for its focused activity.

Reviewer #2 (Recommendations for the authors):1. I object to Aphyd as the name. This is not renaming of an uncharacterized, unstudied generically named gene (i.e., TMEM, KIAA, FAM, cXorfY, etc.), and it is not naming a gene with a phenotype from a genetic screen. ABHD16A is part of the aptly named ABHD family (α/β hydrolase domain). Further, other labs, in numerous publications, have previously studied the phospholipid hydrolase activity of ABHD16A and analyzed phenotypes and mechanisms using KO cells and mice in detail, as appropriately cited by the authors (e.g., Singh et al., Kamat et al.). Making an unneeded (and still pretty vague) extra name for this protein might ultimately confuse people and make it harder to search the literature without adding any extra obvious value. On the contrary, it turns a name that describes a relevant hydrolase fold (ABHD16A) into a neologism that, likely by design, is a homonym of (and conjures up the image of) an insect.

This has been changed back to ABHD16A in the manuscript.

2. Three hours seems like a long time for proximity biotinylation with the optimized TurboID ligase, which is typically used in much shorter pulses (~10 min). The authors are encouraged to comment in the manuscript on the need for such long labeling times.3. What do the "lipase-like motifs" indicated in gray in Figure 1H correspond to? Are they residues H189 and S355 in Figure 6B? The authors should explain more about what they mean by this phrase, as it is confusing that one of them exists in the acyltransferase domain and the other in the α/β hydrolase domain.4. The rescue presented throughout is convincing, but all phenotypes derive from a single ABHD16A siRNA. Inclusion of a second siRNA + rescue for a select subset of experiments would help to increase the rigor of the phenotypic findings.

These points have been addressed above.

5. Line 250: Figure S6A-B callout should be S5A-B.

These points have been addressed above.

6. The data presentation in Figure 3F is confusing, with numbers not adding up (by design but still kind of misleading). I recommend that the authors present it in a different manner, perhaps with n values in the legend and numbers in the graph representing the values in the shaded graph bars? The point is that ABHD16A KD lowers tip-to-mid fusion the most, and rescue can overshoot, but the provided numbers are confusing.

These points have been addressed above.

Reviewer #3 (Recommendations for the authors):I am positive about this study but do have a couple of suggestions.1. It seems unnecessary to rename a protein that has already been studied, especially when the new name doesn't inform on a new function but rather is merely a collection of a different set of letters from the old name. To avoid confusion in the field, the authors may wish to consider keeping the original ABHD16A name.

This has been changed in the manuscript.

2. This initial study conducts rigorous analysis of the role of newly-identified Aphyd in controlling mitochondrial fission/fusion but does not provide any biochemical insights into its lipid-modifying activities – indeed, one can't expect to solve everything in one paper! This question seems complicated since Aphyd contains domains that could catalyze opposite reactions – conversion of a phospholipid such as PS to its lyso- version by a hydrolase activity and conversion of a lyso-phospholipid such as lyso-PS back to a two-chain phospholipid. The analysis of mutants suggests both activities may be needed, but in different ways. In the absence of further biochemical analysis, this last part of the study seems quite speculative to me and could be toned down. Related to this point, in absence of any data showing the lipid specificity of Aphyd activity, the generic fusion/fission model in Figure 8 doesn't add much and the paper would be fine without it.

These points have been addressed above.